# Evolution of an intricate J-protein network driving protein disaggregation in eukaryotes

Nadinath B Nillegoda[1,2]*, Antonia Stank[3,4]†, Duccio Malinverni[5]†, Niels Alberts[1], Anna Szlachcic[1], Alessandro Barducci[6,7], Paolo De Los Rios[5,8], Rebecca C Wade[1,3,9]*, Bernd Bukau[1,2]*

[1]Center for Molecular Biology (ZMBH), Heidelberg University, Heidelberg, Germany; [2]DKFZ-ZMBH Alliance, German Cancer Research Center (DKFZ), Heidelberg, Germany; [3]Heidelberg Institute for Theoretical Studies, Heidelberg, Germany; [4]Heidelberg Graduate School of Mathematical and Computational Methods for the Sciences, University of Heidelberg, Heidelberg, Germany; [5]Laboratory of Statistical Biophysics, School of Basic Sciences, Institute of Physics, École Polytechnique Fédérale de Lausanne, Lausanne, Switzerland; [6]Inserm, U1054, Montpellier, France; [7]CNRS, UMR 5048, Centre de Biochimie Structurale, Université de Montpellier, Montpellier, France; [8]Institute of Bioengineering, School of Life Sciences, École Polytechnique Fédérale de Lausanne, Lausanne, Switzerland; [9]Interdisciplinary Center for Scientific Computing, Heidelberg University, Heidelberg, Germany

**\*For correspondence:**
n.nillegoda@zmbh.uni-heidelberg.
de (NBN); rebecca.wade@h-its.org
(RCW); bukau@zmbh.uni-
heidelberg.de (BB)

†These authors contributed
equally to this work

**Competing interests:** The
authors declare that no
competing interests exist.

**Reviewing editor:** Jeffery W
Kelly, The Scripps Research
Institute, United States

**Abstract** Hsp70 participates in a broad spectrum of protein folding processes extending from nascent chain folding to protein disaggregation. This versatility in function is achieved through a diverse family of J-protein cochaperones that select substrates for Hsp70. Substrate selection is further tuned by transient complexation between different classes of J-proteins, which expands the range of protein aggregates targeted by metazoan Hsp70 for disaggregation. We assessed the prevalence and evolutionary conservation of J-protein complexation and cooperation in disaggregation. We find the emergence of a eukaryote-specific signature for interclass complexation of canonical J-proteins. Consistently, complexes exist in yeast and human cells, but not in bacteria, and correlate with cooperative action in disaggregation in vitro. Signature alterations exclude some J-proteins from networking, which ensures correct J-protein pairing, functional network integrity and J-protein specialization. This fundamental change in J-protein biology during the prokaryote-to-eukaryote transition allows for increased fine-tuning and broadening of Hsp70 function in eukaryotes.

## Introduction

The Hsp70 chaperones are involved in a remarkably broad range of protein folding processes (*Finka et al., 2015*; *Mayer and Bukau, 2005*), which renders them unique among the cellular chaperone systems. This functional versatility is achieved through the activity of an array of cochaperones that regulates the ATP-dependent substrate binding and release cycle of Hsp70 partner chaperones. The members of the J-protein family target Hsp70 to substrates, thereby starting the functional chaperone cycle (*Kampinga and Craig, 2010*). The essential role of J-proteins in diversifying Hsp70 targets and functions is reflected in the expansion of the number of J-protein family members with

**eLife digest** All cells must maintain their proteins in a correctly folded shape to survive. The task of sustaining a healthy set of proteins has increased with the rise of complex life from prokaryotes (such as bacteria) that form simple single-celled organisms to eukaryotes (such as yeast, plants and multicellular animals). As a result of organisms ageing or acquiring genetic mutations, or under stressful conditions such as high temperature, proteins can lose their normal shape and clump together to form "aggregates". These aggregates are potentially toxic to cells and have been linked to many human diseases including neurodegeneration and cancer.

Cells contain molecular machines that help break down aggregates and subsequently recycle or rescue trapped proteins. Some of these machines are based around a protein called Hsp70, which can perform a wide range of protein folding processes. So-called J-proteins help Hsp70 to select aggregates to be targeted for break down. It used to be thought that different classes of J-proteins interacted with Hsp70 separately. However, in 2015, researchers showed that in humans, two different classes of J-proteins can bind to each other to form a "complex", which has distinct aggregate selection properties.

Now, Nillegoda et al. – including several of the researchers involved in the 2015 study – have examined the evolutionary history of these J-protein complexes. This revealed that different classes (A and B) of J-proteins first cooperated after prokaryotes and eukaryotes diverged from each other. In particular, the molecular machinery that breaks down aggregates in yeast cells – but not the machinery found in bacteria – depends on complexes formed from the two classes of J-proteins.

Further investigation revealed that in humans, J-proteins have structural features that ensure they pair up correctly to perform unique activities. Furthermore, Nillegoda et al. suggest that cooperation between J-proteins may have enabled organisms such as humans – which contain over 40 distinct J-proteins – to carry out further specialized protein-folding tasks that do not occur in prokaryotes.

Overall, the findings presented by Nillegoda et al. reveal another important layer to protein quality control in eukaryotic cells. The next step is to understand the possible roles of different J-protein complexes play in J-protein associated cellular protein quality control processes such as preventing protein aggregation, refolding or recycling abnormal proteins. This knowledge could ultimately be used to develop treatments for diseases and disorders in which protein aggregates form.

increasing organismal complexity (*Kampinga and Craig, 2010*; *Nillegoda and Bukau, 2015*). Nucleotide exchange factors (NEFs) reset Hsp70 for the next cycle of substrate binding by stimulating the exchange of ADP with ATP. Additionally, the different types of NEFs may co-determine Hsp70 function by regulating substrate release and communication with downstream protein quality control pathways (*Bracher and Verghese, 2015*).

Traditionally, members of the J-protein family, which is subdivided into classes A, B and C, were viewed as functioning independently, interacting with Hsp70 chaperones in a one-to-one stoichiometry and leading to distinct outcomes in biological processes (*Kampinga and Craig, 2010*). This view has changed with the discovery of complex formation between canonical members of class A and class B J-proteins through transient interactions in metazoa (*Nillegoda et al., 2015*). The domain architecture of the canonical J-proteins consists of the characteristic N-terminal J-domain (JD) linked to the substrate binding C-terminal domain (CTD in class B; Zinc finger-like region (ZFLR)+CTD in class A) via a glycine/phenylalanine (G/F)-rich flexible region. These canonical members dimerize through the C-terminally located dimerization domain (DD; *Figure 1A*) (*Kampinga and Craig, 2010*). The JD is a helical bundle consisting of four α-helices (I, II, III and IV) and a loop region containing the highly conserved tripeptide His-Pro-Asp (HPD) motif (*Figure 1C*). Residues located in α-helices II, III and the HPD motif are implicated in the communication with Hsp70 leading to ATPase stimulation (*Genevaux et al., 2002*). Although structurally related, canonical class A and class B J-proteins show distinct substrate-binding preferences at the CTDs (*Fan et al., 2004*; *Reidy et al., 2014*). The CTDs also provide interaction sites for the JD of opposite class members during

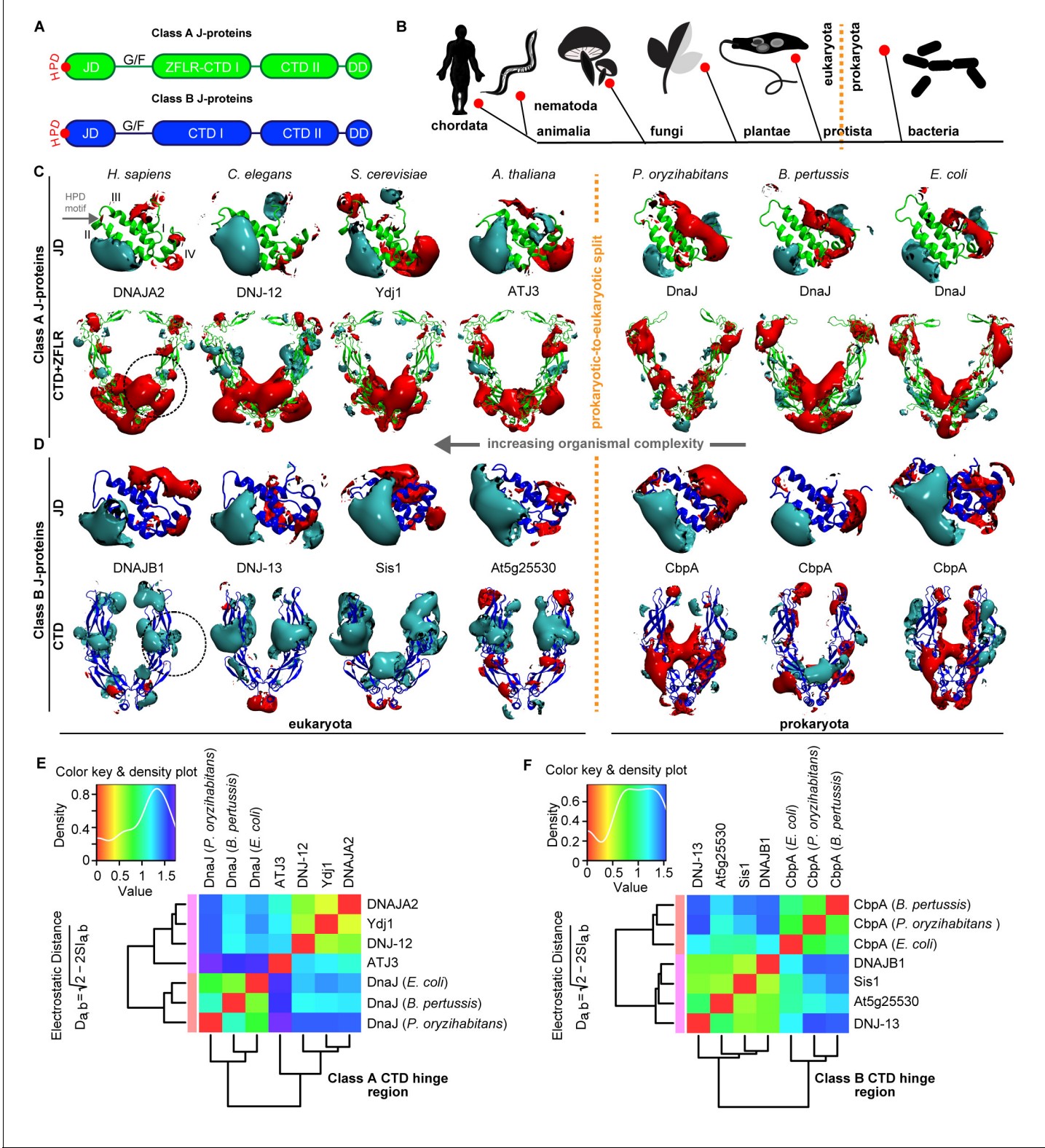

**Figure 1.** Conservation of class-specific electrostatic potential distributions predict interclass J-protein complex formation in eukaryotes. (**A**) Domain architecture of class A and class B J-proteins (shown as protomers). Class A J-proteins have an N-terminal J-domain (JD), a glycine/phenylalanine-rich flexible region (G/F), C-terminal β-sandwich domains (CTD-I and II) and a CTD-I inserted zinc-finger-like region (ZFLR). The Hsp70-interacting HPD motif is indicated in red. Protomer dimerization to form homodimers occurs at the dimerization domain (DD). The ZFLR in CTD-I is absent in the domain architecture of class B J-proteins. (**B**) Evolutionary tree ranking the kingdoms of organisms analyzed in this study. (**C, D**) Electrostatic isopotential

*Figure 1 continued on next page*

*Figure 1 continued*

contour maps (cyan + 1, red −1 kcal/mol/e) of CTD homodimers and of J-domains of class A (green cartoon diagrams) (C) and class B (blue cartoon diagrams) (D) J-proteins. Roman numerals show the four α-helices on class A JD. J-proteins from the following organisms are compared: *Homo sapiens* (DNAJA2, DNAJB1), *Caenorhabditis elegans* (DNJ-12, DNJ-13), *Saccharomyces cerevisiae* (Ydj1, Sis1), *Arabidopsis thaliana* (ATJ3, At5g25530), *Pseudomonas oryzihabitans*, *Bordetella pertussis* and *Escherichia coli* (DnaJ, CbpA). The dashed circles on the CTDs of DNAJA2 and DNAJB1 represent the spherical region used for local PIPSA analysis of electrostatic potential similarity. (E, F) Local PIPSA analysis results for class A CTD (E) and class B CTD (F) electrostatic potentials. The electrostatic potentials in the spherical regions (radius of 25 Å) indicated by the dashed black circles in (C) and (D) were clustered by similarity using Ward's clustering. The heat maps show clustering of J-proteins by similarity (higher similarity indicated by a red shift).

The following figure supplements are available for figure 1:

**Figure supplement 1.** Electrostatic isopotential contour maps of class A J-proteins from humans, fungi, nematodes and bacteria.

**Figure supplement 2.** Electrostatic isopotential contour maps of class B J-proteins from humans, fungi, nematodes and bacteria.

**Figure supplement 3.** Evaluation of JD interaction sites on CTDs of the opposite class J-proteins.

J-protein complex formation (*Nillegoda et al., 2015*). The JD-CTD contact sites display complementary class-specific electrostatic potentials. A negatively charged patch localized mostly on α-helices I and IV of class A and B JDs forms salt bridges with positively charged regions on the CTDs of the opposite class J-proteins (*Nillegoda et al., 2015*) (*Figure 1C,D*). The resulting J-protein complexes have a wider substrate spectrum compared to the individual J-proteins or possible homo J-protein oligomers, because of the amassing of distinct substrate binding modules. This networking strategy is employed by the metazoan Hsp70-based aggregate-solubilizing systems (disaggregases) to broaden the substrate specificity spectrum and to increase machine efficacy (*Nillegoda et al., 2015*).

The gain in protein disaggregation power through interclass J-protein networking gives the human Hsp70-based disaggregase a level of potency comparable to that of the extremely efficient non-metazoan Hsp100-Hsp70 bichaperone disaggregase systems in bacteria, fungi and plants (*Nillegoda and Bukau, 2015*). The Hsp100 (ClpB, Hsp104) component of this bichaperone system, however, disappeared during the evolution of multi-cellular organisms. The discovery of a potent metazoan Hsp70-based disaggregase activity driven by J-protein networking is therefore the missing link in our understanding of efficient amorphous aggregate solubilization in complex organisms. The evolutionary origin of J-protein networking via transient complex formation, however, is unknown. It is also unclear what factors determine and delimit the exact J-protein pairing, particularly within large J-protein families in higher eukaryotes such as humans.

In this study, we investigate the molecular basis for J-protein networking and its evolution. By comparison of structural features of JD and CTD domains of canonical J-proteins across kingdoms of life, we observe a high degree of conservation in electrostatic potentials at the proposed JD and CTD contact faces within each of the classes (A and B) of the eukaryotic J-proteins. Further, phylogenetic and coevolutionary analysis of canonical class A and B J-proteins highlights a distinct phylogenetic signature between prokaryotes and eukaryotes, compatible with interclass J-protein complex formation. Using cell biology and biochemical approaches, we find a switching in J-protein biology at the prokaryote-to-eukaryote transition where class members begin to form functional networks, allowing for the emergence of powerful, yet regulatable eukaryotic disaggregase systems. We, furthermore, decipher the networking code between the cytosolic J-proteins in human cells and describe a naturally occurring strategy to correctly pair interacting J-proteins. This code, based on electrostatic potential distribution patterns at the JD and CTD interaction faces, possibly ensures functional integrity within J-protein networks that expanded during the rise of complex life.

## Results

### Electrostatic potential distributions of CTDs differentiate eukaryotic and prokaryotic J-proteins for interclass complex formation

Although we have previously shown that canonical metazoan J-proteins form interclass complexes with electrostatically complementary interactions between JDs and CTDs, it remains unclear whether these interclass J-protein complexes also occur in non-metazoans, especially given the fact that orthologs of both classes exist in bacteria, protists, fungi, plants and protozoa (*Figure 1B*) (*Nillegoda and Bukau, 2015*). We hypothesized that if interclass J-protein cooperation occurs in these organisms, then structural elements promoting complex formation (*Nillegoda et al., 2015*) must be preserved among canonical class A and class B J-proteins of bacterial and fungal origins.

To investigate our conjecture, we constructed models of the three-dimensional structures of JDs and CTDs of canonical class A and B J-proteins from human to bacteria and investigated the degree of conservation of complementary class-specific electrostatic potentials at intermolecular JD-CTD interfaces. The prokaryotic sample included class A J-protein DnaJ and class B J-protein CbpA from a wide range of bacteria (*Figure 1C,D*). The eukaryotic sample consisted of non-metazoan (plants: ATJ3 (A) and At5g25530 (B); yeast: Ydj1 (A) and Sis1 (B)) and metazoan (nematode: DNJ-12 (A) and DNJ-13 (B); human: DNAJA2 (A) and DNAJB1 (B)) J-proteins belonging to classes A and B. The structures of the protein domains were determined experimentally or modeled by comparative modeling (see Materials and methods). Analysis of JDs showed a general conservation of protein structure and electrostatic potentials within each of the J-protein classes throughout evolution. Both classes A and B JDs displayed a bipolar charge distribution (which is more prominent among class B JDs), where the positive patch around α-helix II implicated in Hsp70 binding was the most prominent feature of the electrostatic potential (*Figure 1C,D*). Among the CTDs, however, clear class-dependent differences were observed between prokaryotic and eukaryotic representatives (*Figure 1C,D*). Qualitatively, the eukaryotic class B$^{CTDs}$ were dominantly positively charged, whereas in prokaryotic structures, we observed a mixture of positively and negatively charged patches (*Figure 1D*). In eukaryotic class A$^{CTDs}$, the ZFLR+CTD-I region is peppered with exposed positively and negatively charged patches, whereas CTD-II was predominantly negatively charged. In contrast, there was a switch of these electrostatic potential patterns in prokaryotic class A J-proteins: the ZFLR+CTD-I regions were dominantly negative, while the CTD-II regions showed clusters of both positive and negative patches (*Figure 1C*).

To quantitatively assess the differences in electrostatic potential among prokaryotic and eukaryotic CTDs, we performed Protein Interaction Property Similarity Analysis (PIPSA) (*Wade et al., 2001*) around the JD interaction interface located at the CTD hinge regions of DNAJA2 and DNAJB1 (*Nillegoda et al., 2015*) (black dotted circles, *Figure 1C,D*; also see Materials and methods). The 25 Å radius spheres encompass residues previously implicated in opposite class JD interaction from crosslinking and Förster resonance energy transfer (FRET) experiments and JD docking simulations (*Nillegoda et al., 2015*). Based on the electrostatic similarities around the hinge regions, the PIPSA analysis showed clustering of the J-proteins into two groups separating the prokaryotes from eukaryotes (*Figure 1E,F*). The J-proteins ATJ3 and At5g25530 from *Arabidopsis thaliana* showed electrostatic potential patterns that were more eukaryotic-like. The clustered groups of CTDs of both classes A and B J-proteins of yeast, nematode and human reflected highly conserved charge distributions at the JD interaction interface (*Figure 1E,F*). The same regions in prokaryotic CTDs showed distinct clustering for both classes but indicated a different charge distribution from the eukaryotic CTDs (*Figure 1E,F*). We conclude that the electrostatically complementary opposite class JD interaction interface is highly conserved among human, worm and yeast J-proteins but not in bacterial counterparts.

We extended the set of bacteria analyzed by modeling additional DnaJ and CbpA CTD structures from Gram-negative proteobacteria (alpha, beta and gamma) and the Gram-positive firmicute *Clostridium ultunense* (*Figure 1—figure supplement 1* and *Figure 1—figure supplement 2*) and made similar observations. However, we observed eukaryotic-like features (e.g. general increase in exposed positive charges at the JD interaction region) emerging in class B$^{CTDs}$ of some bacteria such as *C. ultunense*, *Acetobacter aceti* and *Sphingomonas* sp (*Figure 1—figure supplement 2A*), but not in the partnering class A CTD (*Figure 1—figure supplement 1A* and *Figure 1—figure supplement 3*). Taken together, these features suggest J-protein networking via interclass complex

formation may occur in both animals and simpler eukaryotic unicellular organisms, such as yeast, but not in bacteria.

## Eukaryote-specific phylogenetic signatures for interclass J-protein complex formation

To obtain an in-depth understanding of the evolution of the JD-CTD intermolecular interactions between canonical class A and B J-proteins, we generated class-specific phylogenetic trees and separately analyzed the JD and the CTD regions. The class A and class B trees were built from 12,215 and 4194 sequences, respectively, encompassing all kingdoms of life. As expected, we found the trees to carry coherent phylogenetic signals: eukaryotes (highlighted in gray background) were in general set apart from prokaryotes, and sub-trees were mostly consistent with sub-classifications (*Figure 2A,B* and *Figure 2—figure supplement 1A,B*). Furthermore, eukaryotic class A J-protein sequences of organellar origin were found to mix with sub-trees of prokaryotic regions of the trees (separated by pink lines; *Figure 2B* and *Figure 2—figure supplement 1A*), consistent with their probable bacterial origins (*Lu et al., 2006*; *Deloche et al., 1997*).

We next scanned for eukaryote-specific phylogenetic signatures that supported interclass J-protein complex formation. The scanning was performed with a new approach named Phylogenetic Discriminant Analysis (PDA) that determines the residues that best separate eukaryotes from prokaryotes (see Materials and methods). The PDA analysis applied to JDs of class B (class B$^{JDs}$) identified a total of five positions (*Figure 2C*: residues in red on DNAJB1$^{JD}$ in blue; see also Methods and *Figure 2—figure supplement 2A*) that showed a strong phylogenetic separation between prokaryotes and eukaryotes (eukaryotes highlighted in grey; *Figure 2D*, left). Importantly, of the identified discriminatory hits, two residues mapped onto the E69 and E70 positions on JD of DNAJB1 (*Figure 2C*). These two residues are part of the negatively charged amino acid triplet (D4, E69 and E70 denoted by *) that was previously experimentally identified as having a strong influence in complex formation between the DNAJB1$^{JD}$ and the DNAJA2$^{CTD}$ in humans (*Nillegoda et al., 2015*). The third residue identified (I63) also localized to the same region on α-helix IV of DNAJB1$^{JD}$ (*Figure 2C*). The other residue positions (L29 and K35) flanked the HPD motif (His-Pro-Asp, grey; *Figure 2C*), an essential region for Hsp70 binding (*Suh et al., 1998*; *Jiang et al., 2007*). PDA on class A$^{CTDs}$ highlighted nine positions that strongly discriminated phylogeny between prokaryotes and eukaryotes (*Figure 2C*, *Figure 2D*, right and *Figure 2—figure supplement 2B*). Here, the hits occured in three regions (mapped onto DNAJA2$^{CTD}$ in green cartoon, *Figure 2C*). Of the hits in the CTD, residues Y128 and D222 locate to the CTD-I-CTD-II hinge region. Residue D222 showed the strongest phylogenetic discriminatory value (*Figure 2—figure supplement 2B*). The neighboring residue of D222 (purple; K226, *Figure 2C*) was previously shown to cross-link with the JD of DNAJB1 (*Nillegoda et al., 2015*), placing D222 near the interaction site for class B JD. The hits in the third region were mapped onto the dimerization domain (DD) of class A J-proteins. The phylogenetic discriminatory signal at the DD is currently not understood. To complement the phylogenetic study, we next performed Direct Coupling Analysis (DCA) (*Morcos et al., 2011*) to capture coevolving protein contacts in class B$^{JDs}$ and class A$^{CTDs}$ (see Materials and methods). We observed a statistically significant coevolving residue pair corresponding to V221 on DNAJA2$^{CTD}$ and E62 on DNAJB1$^{JD}$ (orange; *Figure 2C* and *Figure 2—figure supplement 1E*). These residues also map onto the vicinity of the respective JD and CTD interacting regions of DNAJB1 and DNAJA2, further confirming our experimental, structural and PDA-derived findings. Although it is surprising to observe a coevolution between a valine and a glutamic acid, V221 is flanked by a strongly charged region formed by H220, D222 and K223. Further, we also observed in the sequence alignment that the position of V221 is generally surrounded by several charged residues. Thus, V221 may coordinate the local environment of this charged region to interact with charged residues proximal to E62 on DNAJB1$^{JD}$. The overlap of PDA hits with the DCA and experimentally implicated residues in interclass J-protein complex formation implies the presence of a eukaryote-specific phylogenic signature for the interaction between class B$^{JDs}$ at the CTD hinge region of class A$^{CTDs}$.

A reciprocal phylogenetic analysis was next performed with class A$^{JDs}$ and class B$^{CTDs}$, which provided similar observations (*Figure 2—figure supplement 1* and *Figure 2—figure supplement 2C, D*). The positions that best capture prokaryotic to eukaryotic phylogenetic splitting on class B$^{CTDs}$ showed two clusters localized to either CTD or DD regions (*Figure 2—figure supplement 1C,D*). Importantly, the CTD cluster containing residues I175 and K209 were located at the hinge region

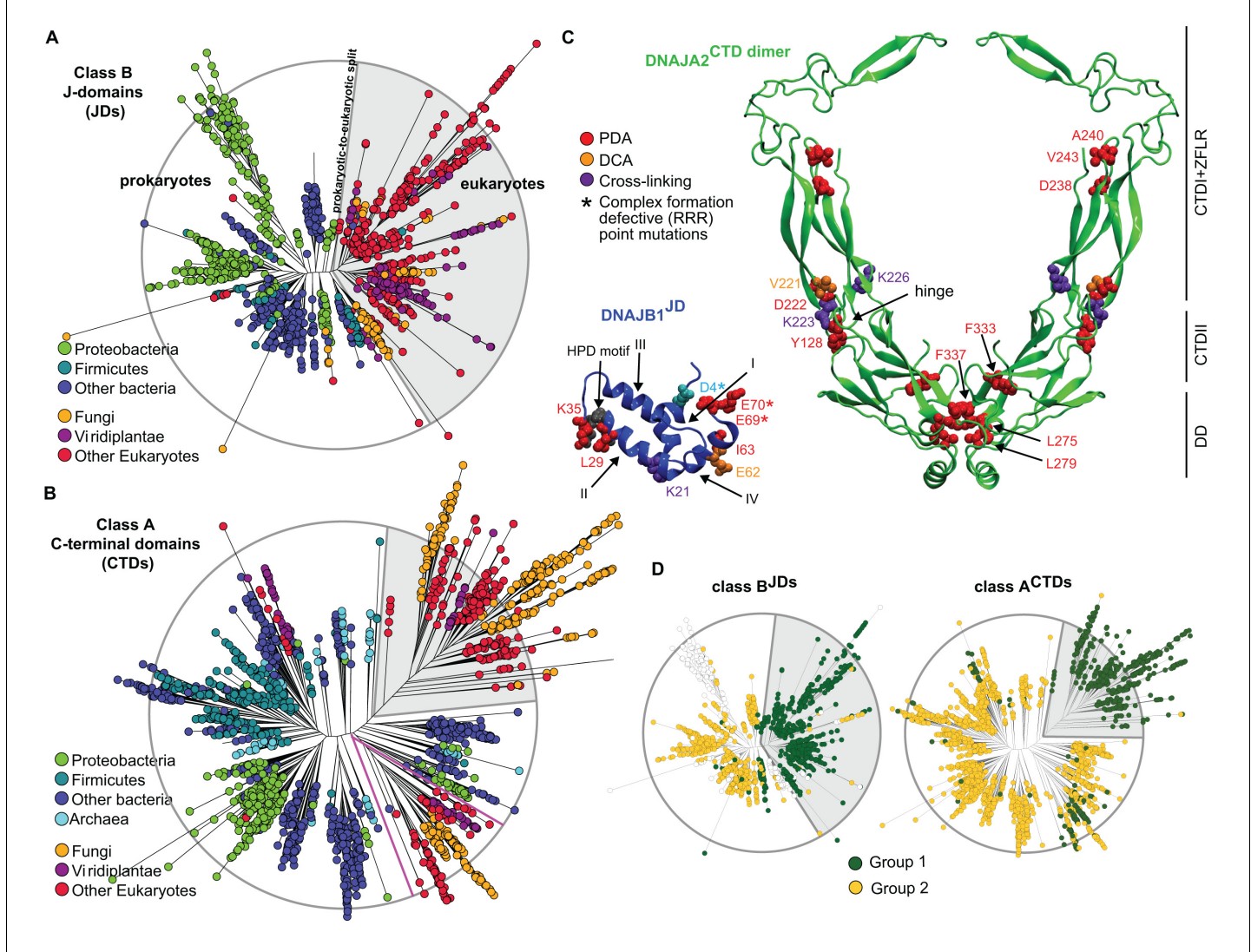

**Figure 2.** Phylogenetic and coevolutionary analyses of the JD-CTD interaction between class A and B J-proteins. (A) Phylogenetic tree of class B J-domains. Color-coding separates different phylogenetic groups. Grey area highlights the separation of eukaryotes (fungi, viridiplantae and other eukaryotes) from prokaryotes (proteobacteria, firmicutes and other bacteria) and Archaea. (B) As in (A), phylogenetic tree of class A CTDs. Pink lines delimit organellar sequences of eukaryotic organisms. (C) Structural view of most discriminating positions predicted by PDA (red) plotted on JD of DNAJB1 (blue, five positions) and CTD of DNAJA2 (green, nine positions). DCA-derived coevolving residue pairs depicted on DNAJB1<sup>JD</sup> and DNAJA2<sup>CTD</sup> (orange). Experimentally determined cross-linking residues between DNAJB1 and the DNAJA2 are indicated in purple (***Nillegoda et al., 2015***). Location of the triple charge reversion (E/D→R) mutations that disrupts interclass J-protein complex formation between DNAJB1 and DNAJA2 denoted by (*) (***Nillegoda et al., 2015***). The HPD motif of DNAJB1<sup>JD</sup> is shown in grey. Roman numerals show the four α-helices on JD. (D) Mapping of sequence clustering derived from PDA (see Materials and methods) using the most discriminating positions on to JD and CTD trees of class B and class A J-proteins, respectively. The two identified groups (green and yellow nodes) covered 81% in the case of the clustering done on the JDs and 100% when clustering was done on the CTDs. Unclassified sequences are depicted in white.

The following figure supplements are available for figure 2:

**Figure supplement 1.** Phylogenetic and coevolutionary analyses between class A<sup>JDs</sup> and class B<sup>CTDs</sup>.

**Figure supplement 2.** Evaluation of robustness in Phylogenetic Discriminant Analysis.

(mapped on the DNAJB1$^{CTD}$ structure). K209 residue was previously observed to cross-link with the JD of DNAJA2 (*Nillegoda et al., 2015*). Class A JDs, however, provided less clear-cut results. Nevertheless, DCA captured a coevolving residue at position R63, which is flanked by E61 and E64 (*Figure 2—figure supplement 1C,F*). These residues form the class B CTD interaction interface on the JD of DNAJA2 (*Nillegoda et al., 2015*). The coevolving G278 residue localized to the JD interaction interface in the DNAJB1$^{CTD}$ hinge region.

In summary, we find phylogenetic and coevolutionary signatures compatible with bi-directional JD-CTD interactions between class A and class B J-proteins of eukaryotic origins. These findings support the evolution of interclass J-protein networks beyond metazoa but not in prokaryota. Our data show the emergence of two protein-protein interaction regions in eukaryotic JDs: One for Hsp70 and the second for interclass J-protein complex formation. We also identified a similar region for partner protein interaction at the CTDs of canonical J-proteins. When combined, PDA and DCA analyses identify the hinge region between CTD-I and II subdomains as the primary interface for JD binding during interclass J-protein complex formation, which agrees with our previous crosslinking, FRET and docking simulation results obtained with human J-proteins (*Nillegoda et al., 2015*). The recently reported interaction between ubiquitin ligase Rsp5 and Ydj1 for targeting aberrant proteins during heat stress also highlights the importance of this hinge region for partner protein binding (*Fang et al., 2014*). This protein-protein interaction interface shows no significant overlap with known substrate-binding regions of canonical J-proteins. Consistently, binding of JD fragments to CTDs does not block substrate association with J-proteins (*Nillegoda et al., 2015*).

## In situ analysis of interclass J-protein complex formation in bacteria, yeast and human cells

To validate our structural, phylogenetic and coevolutionary analyses, we employed an in situ antibody-based proximity ligation assay (PLA) (*Söderberg et al., 2006*) to visualize interclass J-protein complex formation in cells. J-protein complexes containing human class A DNAJA2 and class B DNAJB1 were detected as red puncta after signal amplification (each punctum representing a complex formation event) in cultured human HeLa cells (*Figure 3A*). This reconfirms our original biochemical findings for interclass J-protein complex formation in humans (*Nillegoda et al., 2015*). Addition of either one of the J-protein specific antibodies alone did not generate red puncta (*Figure 3B,C*). To further validate the specificity of the interactions, we prevented J-protein pairing by separately depleting each member using RNAi knockdowns. As expected, we observed a drastic decrease in the number of complexes per cell in the individual J-protein depletions (*Figure 3D–F*). The J-protein knockdowns and antibody specificities were confirmed by Western blotting (*Figure 3—figure supplement 1A*). Next, we performed the same assay in *S. cerevisiae* cells for the two major yeast cytosolic J-proteins, class A Ydj1 and class B Sis1. We observed a strong punctated red fluorescence signal in wild type cells, confirming the predicted complex formation between the two yeast J-proteins (*Figure 3G*). Control cells carrying a deletion of the *ydj1* gene showed no red puncta (*Figure 3J* and *Figure 3—figure supplement 2A*). Similarly, depletion of the essential Sis1 (expressed from TetO7 repressible promoter) using doxycycline (+dox) also showed a dramatic decrease in complex formation in cells (*Figure 3K,L* and *Figure 3—figure supplement 2B*). Non-specific signal amplification was not observed in controls lacking either one of the J-protein antibodies (*Figure 3H,I* and *Figure 3—figure supplement 2C*).

To discern complex formation between DnaJ and CbpA, the only class A and B J-protein members in *E. coli*, we also reconstituted the proximity assay in bacterial cells. In contrast to the interclass J-protein interactions observed in eukaryotic cells, we did not observe complexes between DnaJ and CbpA after signal amplification in log phase *E. coli* cells (*Figure 3M*) where both J-proteins are expressed (*Figure 3—figure supplement 2D*) (*Tatsuta et al., 1998*). Cellular CbpA levels reach maximum levels during stationary phase in response to nutrition starvation (*Yamashino et al., 1994*), but no interclass DnaJ-CbpA complexes were observed in *E. coli* cells grown into stationary phase (*Figure 3—figure supplement 2D,E*). In contrast, our positive controls captured JD-driven interactions between DnaJ and DnaK (bacterial Hsp70) and CbpA and DnaK (*Figure 3N* and *Figure 3—figure supplement 2F*). C-terminal YFP and mCherry tagging of DnaJ and CbpA does not compromise in vivo functions, JD-mediated interactions or localization (*Winkler et al., 2010*; *Chintakayala et al., 2015*). These findings corroborate the structural data that showed absence of complementary JD interaction interfaces on prokaryotic CTDs (*Figure 1C,D*). In essence, we now provide direct

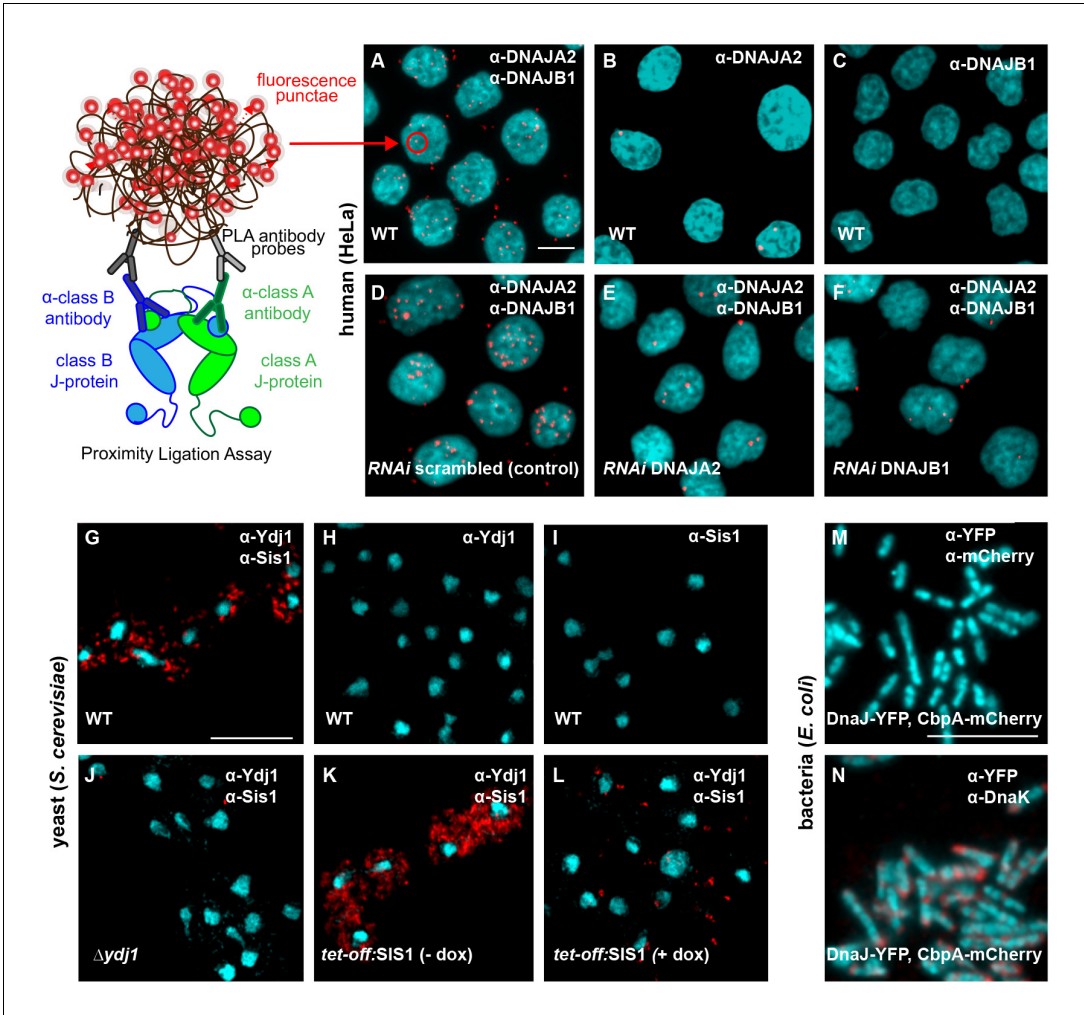

**Figure 3.** Interclass J-protein complexes occur in human and yeast cells, but not in bacteria. (**A–F**) Detection of interclass J-protein complexes in human cells. (**A**) DNAJA2 (class A) and DNAJB1 (class B) form interclass complexes in HeLa cells. Red punctae reflect fluorescent signal amplification from single DNAJA2+DNAJB1 interaction events using an in situ proximity ligation assay. Fluorescently labeled oligonucleotides (red) hybridize to amplified the DNA proximity signal (see cartoon on left). Nuclei stained with DAPI (cyan). (**B, C**) Technical controls of PLA for antibody specificity. (**B**) PLA performed with anti-DNAJA2 antibody only. (**C**) PLA performed with anti-DNAJB1 antibody only. (**D–F**) RNAi knockdown of either DNAJA2 (**E**) or DNAJB1 (**F**) disrupts interclass J-protein complex formation in HeLa cells. Control reaction with scrambled RNAi shown in (**D**). (**G–L**) Interclass J-protein complex formation in yeast (*Saccharomyces cerevisiae*) cells. (**G**) Appearance of red punctae denotes complex formation between Ydj1 (class A) and Sis1 (class B). Nuclei stained with DAPI (cyan). (**H**) PLA performed with anti-Ydj1 antibody only. (**I**) PLA performed with anti-Sis1 antibody only. (**J**) Ydj1-Sis1 complex formation is abrogated in cells with Ydj1 gene knocked out. (**K, L**) PLA against Ydj1 and Sis1 in S. *cerevisiae* cells depleted of Sis1. Sis1 expression is switched off (*tet-off*) in the absence (**K**) or presence (**L**) of doxycycline (see Methods). (**M**) No complexing events occur between DnaJ-YFP (class A) and CbpA-mCherry (class B) in *Escherichia coli* log phase cells after signal amplification. Primary antibodies directed toward YFP and mCherry tags. Bacterial DNA stained with DAPI (cyan). (**N**) Complex formation detected between DnaJ-YFP (class A) and DnaK in log phase *E. coli* cells. Primary antibodies target YFP tag and DnaK. Scale bar = 10 μm. n (biological repeats) = 3.

The following figure supplements are available for figure 3:

**Figure supplement 1.** J-protein levels after knockdown or overexpression in HeLa cells.

**Figure supplement 2.** Yeast and bacterial class A and class B J-protein levels analyzed by western blotting.

evidence for interclass J-protein complex formation in both metazoan and non-metazoan eukaryotic cells, which seems to be absent in bacteria.

## Interclass J-protein complex formation potentiates yeast, but not bacterial Hsp100-Hsp70 bichaperone disaggregase systems

Previously, we showed that the Hsp70-based protein disaggregases in nematodes and humans rely on complex formation between class A and B J-proteins to expand substrate recognition and potentiate solubilization of amorphous aggregates (*Nillegoda et al., 2015*). As Ydj1 and Sis1 also form interclass J-protein complexes in *S. cerevisiae* cells, we assessed the impact of this phenomenon on the function of the yeast protein disaggregase system (*Figure 4*). As a control, we used the homologous disaggregase system from *E. coli*, where interclass communication between J-proteins is apparently absent. The aggregate solubilizing bichaperone machines in bacteria and yeast critically depend on the cooperation between the powerful Hsp100 AAA+ ATPases and the Hsp70 system (*Glover and Lindquist, 1998*; *Goloubinoff et al., 1999*; *Kaimal et al., 2017*). Hsp70 recruits and activates Hsp100 while the J-proteins provide the overall substrate selectivity for the disaggregase system (*Mogk et al., 2015*). Despite considerable similarities in machine architecture and function (*Mogk et al., 2015*; *Sousa, 2014*), the bacterial ClpB-DnaK bichaperone system is separated from the yeast counterpart (Hsp104-Ssa1) by more than 2 billion years of evolution in which the prokaryote-to-eukaryote transition occurred (*Hedges et al., 2004*).

In vitro disaggregation/refolding reactions containing the yeast bichaperone system with either Ydj1 (A, green) or Sis1 (B, blue) showed considerable reactivation of preformed aggregates of model substrate firefly luciferase at high chaperone to substrate ratios (*Figure 4—figure supplement 1A*) consistent with previous reports (*Seyffer et al., 2012*; *Glover and Lindquist, 1998*). Of note, the amorphous luciferase aggregates used in these assays were generated in the presence of a small heat shock protein (Hsp26) to mimic in vivo protein aggregation conditions (*Cashikar et al., 2005*; *Haslbeck et al., 2005*) and allow increased substrate accessibility for disaggregation machineries (*Rampelt et al., 2012*; *Nillegoda et al., 2015*). Compared to single J-protein-containing reactions (class A, green; class B, blue), the reactions consisting of both Ydj1 and Sis1 (magenta) showed synergistic increase in luciferase reactivation, especially under non-saturating conditions (*Figure 4B* (non-saturating) and *Figure 4—figure supplement 1A* (saturating)) similar to the human HSPA8-HSPH2 (Hsp70-Hsp110) disaggregase system containing DNAJA2 (A) and DNAJB1 (B) under identical experimental conditions. We conclude that the yeast disaggregase system is geared to function with both Ydj1 and Sis1 individually, but requires synergistic action of the two J-proteins to gain high efficiency similar to metazoan Hsp70-based disaggregases. In contrast, the refolding-only reactions performed with soluble misfolded monomeric luciferase (*Figure 4A*, see scheme) using yeast Hsp70 system (Ssa1-Sse1-Ydj1/Sis1) showed no synergy in the presence of class A + class B J-proteins (*Figure 4—figure supplement 1B*). Therefore, the observed synergy in aggregate resolution occurs at the upstream disaggregation step and not in the subsequent polypeptide refolding step mediated solely by the yeast Hsp70 system. On the contrary, under similar conditions, the *E. coli* bichaperone disaggregase system reactivated aggregated luciferase very efficiently independent of J-protein class and mixing (*Figure 4C*). The *E. coli* Hsp70 (DnaK) system also did not require the cooperation between DnaJ and CbpA for efficient refolding of soluble misfolded luciferase (*Figure 4—figure supplement 1C*). Collectively, our functional assays show eukaryotic disaggregases are cogged to depend on J-protein complex formation for efficacy, which is absent in bacteria. These findings are fully consistent with our biological, phylogenetic, coevolutionary and protein structure based observations.

## J-protein class-specific aggregate targeting regulates yeast disaggregase system

To dissect the disaggregation process further and determine aggregate specificities of bacterial and yeast class A and B J-proteins, we employed a size-exclusion chromatography (SEC)-based strategy (*Nillegoda et al., 2015*). SEC in principle separates proteins (monomeric/oligomeric) in solution by size. However, this separation may be influenced not only by the size but also by the shape and density (nature of packing) of the protein aggregates detected. Tritiated ($^3$H) luciferase after heat denaturation/aggregation elutes in two peaks (*Figure 4D*) (*Nillegoda et al., 2015*). For ease of analysis,

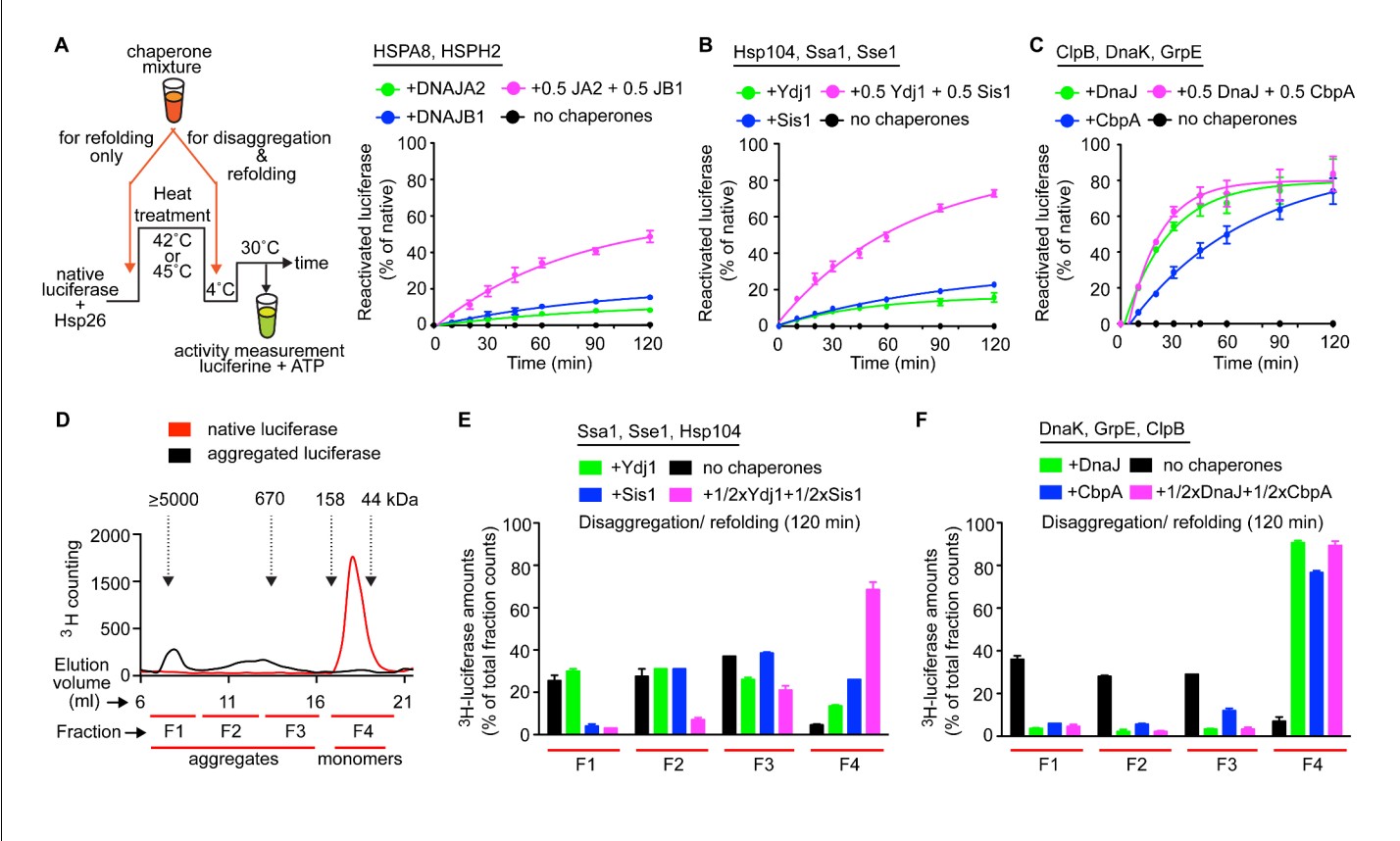

**Figure 4.** Interclass J-protein communication has distinct functional consequences for prokaryotic and eukaryotic protein disaggregation systems. (**A**) Scheme for in vitro disaggregation/refolding and refolding-only reactions. Solubilization of preformed heat-aggregated luciferase using the human HSPA8-HSPH2 (Hsp70-Hsp110) system with DNAJA2 (Class A, green), DNAJB1 (Class B, blue) or DNAJA2+DNAJB1 (magenta). Control reaction containing aggregates without chaperone mix (black). n = 3. (**B**) As in (**A**), reactivation of luciferase aggregates by the yeast Hsp104-Ssa1 (Hsp100-Hsp70) bichaperone disaggregation system with J-proteins Ydj1 (class A, green), Sis1 (Class B, blue) and Ydj1+Sis1 (magenta). The NEF is Sse1, which is homologous to HSPH2. n = 3. (**C**) Reactivation of aggregated luciferase by the bacterial ClpB-DnaK (Hsp100-Hsp70) bichaperone system with GrpE (NEF) in the presence of DnaJ (Class A, green), CbpA (Class B, blue) or DnaJ+CbpA (magenta). (**D**) SEC profile of aggregated [3]H-luciferase (elution fractions F1-F3, black). Soluble luciferase monomers (~63 kDa) elute in fraction F4 (red). MW scale on top in kDa. (**E,F**) Aggregate quantification for F1-F4 in SEC profile obtained with yeast (**E**) and bacterial (**F**) bichaperone systems after 120 min of disaggregation. Profiles show the disappearance of [3]H-luciferase from aggregates (F1-F3) with concomitant accumulation of the disaggregated monomers in F4. Black bars indicate aggregate levels obtained from control reactions lacking chaperones. Values normalized to total counts in each reaction. n = 3. Error bars reflect mean ± sem. See Materials and methods for protein concentrations used.

The following figure supplements are available for figure 4:

**Figure supplement 1.** Yeast class A and B J-proteins form interclass complexes to synergize disaggregation activity of the Hsp104-Ssa1 bichaperone system.

**Figure supplement 2.** Cross species communication between class A and B J-proteins in protein refolding and protein disaggregation.

we further divided the broad aggregate elution peak into two fractions and defined a total of three aggregate fractions (F1, F2 and F3). For simplicity, we define these aggregate fractions by size (F1, ≥5000 kDa; F2, 700–4000 kDa; F3, 200–700 kDa) (*Figure 4D*). Soluble monomeric luciferase (~63 kDa) elutes in the F4 fraction. Aggregate resolution was quantified by the degree of tritium counts remaining in each fraction compared to control (black) post disaggregation. Using this aggregate profiling system, we observed that Ydj1 (A, green) and Sis1 (B, blue) J-proteins alone selectively target the yeast Hsp100-Hsp70 disaggregase to resolve small (F3) and large (F1) luciferase aggregates, respectively (*Figure 4E*). However, when combined, the resulting interclass J-protein

complexes targeted the disaggregase machinery to all aggregate populations, including medium-sized aggregates eluting in F2 fraction (*Figure 4E* and *Figure 4—figure supplement 1D*). In accordance, a synergistic increase in disaggregated luciferase monomers appeared in fraction F4 (magenta, *Figure 4E*). The human disaggregase system, containing DNAJA2 (A) and DNAJB1 (B), also showed similar aggregate resolution patterns (*Nillegoda et al., 2015*), indicating interclass J-protein action in broadening aggregate targeting specificity is conserved from yeast to human.

In the absence of Hsp104, the yeast Hsp70 system showed relatively low luciferase aggregate resolubilization at 120 min (*Figure 4—figure supplement 1E*) consistent with in vivo findings (*Hsieh et al., 2014*). However, residual disaggregase activity was observed with detectable drops in F1-F3 aggregate levels in the mixed class J-protein containing reaction, although insufficient for full J-protein cooperation as observed when Ssa1 was substituted with human HSPA8 (*Figure 4—figure supplement 1F*). These findings possibly reflect an early stage of the evolution of Hsp70-based disaggregase systems that appear powerful in metazoa (*Nillegoda and Bukau, 2015*).

## *E. coli* J-proteins target ClpB-DnaK disaggregase to all aggregate populations independent of J-protein class

Our earlier findings showed that the ClpB-DnaK bichaperone system reactivates aggregated luciferase with high efficiency independent of J-protein class (*Figure 4C*) and complex formation (*Figure 3M*). To gain mechanistic insight, we examined how the *E. coli* disaggregase system achieves this robustness in the absence of interclass J-protein complex formation. The SEC profiling assays showed that DnaJ and CbpA lacked class-based selective aggregate targeting. Both bacterial J-proteins independently guided the ClpB-DnaK disaggregase to all aggregate sizes in F1-F3 fractions (*Figure 4F*), consistent with the luciferase activity results in *Figure 4C*. The shorter reaction time point (40 min), however, showed the appearance of class B-like specialization in CbpA revealed by a delay in solubilizing smaller aggregates in F2-F3 (blue, *Figure 4—figure supplement 1G*). These smaller aggregates were, nevertheless, completely resolved at 120 min (*Figure 4F*). Subtle synergistic variations in activities masked by robustness of the bacterial system may yet hint toward some degree of J-protein cooperation. We, however, did not observe such synergistic activity even when the bacterial disaggregase components were further depleted by three fold to limit activity (*Figure 4—figure supplement 1H*). Unbiased Brownian dynamics docking simulations (*Martinez et al., 2015*) between JD and CTD of CbpA and DnaJ did not capture the preferred binding arrangement described for interclass complex formation between human J-proteins (*Nillegoda et al., 2015*) (*Figure 4—figure supplement 1J*), further corroborating our structural, biochemical and cell biological findings. Reactions containing only the bacterial Hsp70 system showed considerably less aggregate dissolution at 120 min (*Figure 4—figure supplement 1I*). In essence, based on this example, bacteria use a different strategy to achieve high efficiency in protein disaggregation, evading a requirement for interclass J-protein complex formation. The extremely high overlap between DnaJ and CbpA for luciferase aggregate selection may explain the simple interchangeability of these two J-proteins for disaggregase targeting function in vivo (*Winkler et al., 2012*). Overall, we observe key operational changes between bacterial and yeast bichaperone disaggregase systems mediated by different J-protein configurations.

## Cross-species chaperone communication reveals additional specialization of eukaryotic J-proteins for Hsp70-based protein disaggregation

Our data show that class-specific specialization among eukaryotic J-proteins restrict the aggregate sizes that can be solubilized by the Hsp70 system. We reasoned that the less specialized class A or class B bacterial J-proteins maybe able to override this selectivity when combined with the human Hsp70-based disaggregase, leading to increase of disaggregation activity. We first assessed the cross-species communication between bacterial J-proteins and human Hsp70 system. Both bacterial DnaJ and CbpA cooperated with human HSPA8 and HSPH2 to refold heat-denatured, but predominantly monomeric luciferase (*Nillegoda et al., 2015*) (*Figure 4A* scheme, *Figure 4—figure supplement 2A*). This is consistent with stimulation of ATP hydrolysis in HSPA8 by bacterial J-proteins (*Minami et al., 1996*). Of note, the DnaJ-containing reaction showed a considerably increased luciferase activity compared to CbpA-containing reactions, consistent with the higher refolding

capacity of class A J-proteins with Hsp70 (*Nillegoda et al., 2015*). However, deferring from the cross-species collaboration observed for protein refolding and contrary to our expectations, DnaJ and CbpA (single or mixed) completely failed to cooperate with the human Hsp70 system for protein disaggregation (*Figure 4—figure supplement 2B*). In contrast, yeast Ydj1 and Sis1 fully complemented the human orthologs in both protein refolding and disaggregation. Further, when paired with human DNAJA2 and DNAJB1 to form A+B cross-species J-protein combinations, only yeast (and not *E. coli*) members could cooperate and synergize protein disaggregation (*Figure 4—figure supplement 2B*) well above additive effects (*Figure 4—figure supplement 2C*).

To further validate these observations, we next tested for cross-species physical interaction between human and yeast class A and B J-proteins in vitro using a previously established FRET-based competition assay, which captures the intermolecular JD-CTD bidirectional cross interaction (indicated by red dotted lines, *Figure 4—figure supplement 2D*) between human DNAJA2 and DNAJB1 (*Nillegoda et al., 2015*). Addition of five-fold excess unlabeled DNAJA2 or DNAJB1 decreased energy transfer (measured as donor quenching) between fluorophores located at the JD of DNAJA2 (labeled with acceptor fluorophore ReAsH attached to the CCGPCC motif at the N-terminus of the JD) and the hinge region of the CTD of DNAJB1 (labeled with donor fluorophore Alexa Fluor 488 at residue Cys278) (*Figure 4—figure supplement 2D*) consistent with our previous findings (*Nillegoda et al., 2015*). Similarly, addition of excess unlabeled yeast, but not bacterial J-proteins competed with complex formation between the labeled human class A and B J-proteins (*Figure 4—figure supplement 2D*). The observed cross-species interactions among yeast and human J-proteins correlate with the general conservation of electrostatic potentials at the JD-CTD interaction sites on eukaryotic class A and B J-proteins (*Figure 1C,D*). The level of donor quenching by the yeast J-proteins was however less compared to the human homologs. This is likely due to the composite nature of J-protein complex formation and small structural variations at the JD-CTD interaction interfaces (*Figure 1E,F*). Together, these findings suggest that the communication between class A and class B J-proteins is not only playing a role in aggregate selection, but also important for Hsp70-based disaggregase machine assembly and/or architecture.

## 'Deal-breakers' show how correct J-protein pairing occurs to achieve functional integrity in eukaryotic J-protein networks

The number of class A and B members in the cytosol have multiplied during evolution from prokaryotes to eukaryotes (*Nillegoda and Bukau, 2015*; *Kampinga and Craig, 2010*). The interclass J-protein networking function we describe provides additional flexibility to eukaryotic Hsp70 systems in extending the range of targeted substrates and protein quality control processes. This raises the question how prevalently interclass J-protein networking is employed for distinct functions such as disaggregation within expanded J-protein families of eukaryotes. The human cytosol contains four canonical class A members (DNAJA1, DNAJA2, DNAJA4 and an isoform of the mitochrondrial DNAJA3) and nine class B J-proteins of which DNAJB1, DNAJB4 and DNAJB5 constitute the canonical members. The rest of class B members form a set of non-canonical J-proteins that possess an N-terminal JD and a G/F rich region, but lack the two $\beta$-sheet rich barrel topology CTDs and the DD (*Lu et al., 2006*; *Kampinga and Craig, 2010*). Instead, these non-canonical class B J-proteins, such as DNAJB2 and DNAJB8, have very diverse C-terminal domains with distinct functions (*Kampinga and Craig, 2010*). DNAJB2 and DNAJB8, but not DNAJB1, specialize in preventing aggregation of amyloidogenic proteins via the C-terminally located ubiquitin-interacting motifs (UIMs) and serine-rich stretches (SSF-SST) (*Figure 5A*), respectively (*Labbadia et al., 2012*; *Hageman et al., 2010*; *Gillis et al., 2013*). Moreover, DNAJB2 seems to compete with the folding/holding chaperone activities of DNAJB1 and to enhance the degradation of misfolded proteins prior to their aggregation in cells (*Howarth et al., 2007*; *Westhoff et al., 2005*). DNAJB8 functions as a poly-dispersed oligomeric complex instead of forming dimers as observed with DNAJB1.

We further checked for functional divergence within these class B J-protein family members by assessing the cooperation of the two non-canonical members with DNAJA2 (class A, canonical member) in protein disaggregation. Compared to DNAJB1, DNAJB2 and DNAJB8 were incapable of reactivating aggregated luciferase even when mixed with DNAJA2 (*Figure 5B*), showing a clear separation in function. Moreover, FRET competition assays revealed that these specialized J-proteins were unable to complex with DNAJA2 and DNAJB1 (*Figure 5C*).

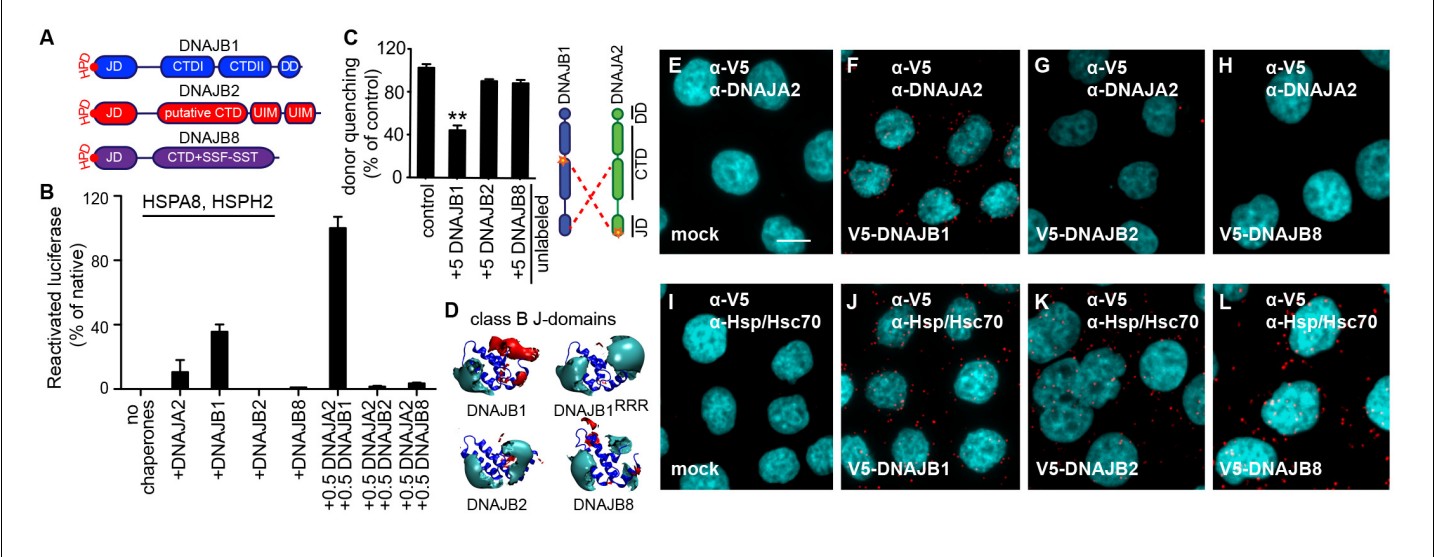

**Figure 5.** A naturally occurring discriminatory strategy correctly pairs J-proteins for specialized functions in eukaryotes. (**A**) Structural organization of canonical (DNAJB1) and non-canonical (DNAJB2 and DNAJB8) members of the class B J-protein subfamily. (**B**) Reactivation of aggregated luciferase by the human Hsp70-based disaggregase system with canonical and non-canonical J-proteins under saturating chaperone levels. n = 3. See Materials and methods for protein concentrations used. (**C**) Competition of unlabeled DNAJB1, DNAJB2 and DNAJB8 for the JD-CTD interaction between DNAJB1 and DNAJA2 analyzed by FRET. Bars represent donor quenching efficiency of JD and CTD intermolecular interactions. Cartoon shows fluorophore positions mapped onto DNAJB1 and DNAJA2 protomers. Red dotted lines indicate intermolecular JD-CTD bidirectional interactions. The N-terminus of DNAJA2$^{JD}$ and the C-terminus of DNAJB1$^{CTD}$ (at residue Cys278) were labeled with the acceptor fluorophore ReAsH and the donor fluorophore Alexa Fluor 488, respectively. n = 3. Two-tailed t-test **p<0.01. (**D**) Electrostatic isopotential contour maps (cyan + 1, red −1 kcal/mol/e) of the J-domains of DNAJB1, DNAJB2 and DNAJB8. DNAJB1$^{RRR}$ is the triple mutant of DNAJB1 (D4R, E69R, E70R) that fails to interact with opposite class CTDs (*Nillegoda et al., 2015*). (**E–L**) Interclass interactions between V5-DNAJB1 (control), V5-DNAJB2 and V5-DNAJB8 with either DNAJA2 (**E–H**) or Hsp70 (**I–L**) captured by PLA in HeLa cells. Red punctae reflect a single complex formation event between the indicated J-proteins. PLA controls for antibody specificities in mock transfected cells (**E, I**). Nuclei stained with DAPI (cyan). Scale bar = 10 μm. n (biological repeats) = 2. The Hsp70 antibody recognizes both human Hsp70 and Hsc70 variants.

The following figure supplement is available for figure 5:

**Figure supplement 1.** The J-domain of DNAJB8 is insufficient for interclass J-protein cooperation and disaggregation synergy.

How is this functional and physical separation, which seems crucial to avoid futile interactions among different J-protein class members effectively established? The minimal structural element required for interclass J-protein communication is the JD. Isolated JDs effectively bind to opposite class CTDs and compete with complex formation between wild-type J-proteins (*Nillegoda et al., 2015*). Therefore, we hypothesized that JDs of DNAJB2 and DNAJB8 must contain inherent signals to prevent interclass interactions. Since JD-CTD interactions are sensitive to high-salt concentrations (*Nillegoda et al., 2015*), we examined the electrostatic potentials at the JDs of these non-canonical members. In JDs of DNAJB2 and DNAJB8, the negatively charged patch critical for opposite class CTD interaction was replaced by a highly positively charged region (*Figure 5D*). This configuration was similar to that of the previously analyzed complex-forming defective, triple charge-reversal variants (RRR) of class A and B J-proteins (*Nillegoda et al., 2015*) (e.g. DNAJB1$^{RRR}$; D4R, E69R and E70R (RRR), *Figure 5D*). We could also quantitatively show that the electrostatic potentials in the region around helices I and IV of the JDs of DNAJB2, DNAJB8 and DNAJB1$^{RRR}$ clearly differ from the of the electrostatic potentials of the same region in canonical DNAJB1/DNAJB4 JDs (*Figure 5— figure supplement 1A*). We presumed that DNAJB2 and DNAJB8 avoided interaction with DNAJA2 using this naturally occurring charge reversal (negative to positive) at the JDs as a repulsive signal.

To test our presumption, we generated a DNAJB1 chimera containing either the JD of DNAJB8 or the JD of CbpA (control). The *E. coli* CbpA$^{JD}$ has a charge bipolarity character comparable to eukaryotic class B$^{JDs}$ (*Figure 1D* and *Figure 5—figure supplement 1A*). FRET competition assays

with excess unlabeled DNAJB8$^{JD}$-DNAJB1$^{CTD}$ and CbpA$^{JD}$-DNAJB1$^{CTD}$ chimeras showed equal reduction in donor quenching, but to a lesser degree compared to wild-type DNAJB1 (*Figure 5—figure supplement 1B*). The partial FRET donor quenching by the DNAJB8$^{JD}$-DNAJB1$^{CTD}$ chimera most likely resulted from a unidirectional JD–CTD interaction in which the JD of DNAJA2 interacts with the CTD of DNAJB1 in the DNAJB8$^{JD}$-DNAJB1$^{CTD}$ chimera. Consistently, this partial intermolecular tethering was completely abolished when the FRET experiment was repeated with an N-terminally ReAsH labeled DNAJA2$^{RRR}$ (*Nillegoda et al., 2015*) (*Figure 5—figure supplement 1C*). DNAJA2$^{RRR}$ carries a J-domain charge reversal triple mutation (D6R, E61R and E64R), which diminishes its JD from binding to the CTD of the DNAJB8$^{JD}$-DNAJB1$^{CTD}$ chimera. As a negative control, we used unlabeled DNAJB1$^{RRR}$ as a competitor, and as expected, this complex forming defective mutant behaved similar to the DNAJB8$^{JD}$-DNAJB1$^{CTD}$ chimera (*Figure 5D* and *Figure 5—figure supplement 1C*). Importantly, the degree of intermolecular tethering by a unidirectional JD–CTD interaction was insufficient for full J-protein cooperation and disaggregation efficiency (*Figure 5—figure supplement 1D–F*), consistent with previous observations (*Nillegoda et al., 2015*). We surmise that a bidirection JD–CTD tethering may stabilize and/or contribute to a specific architecture for J-protein oligomerization that facilitates the assembly of the Hsp70-based disaggregase on the surface of an aggregate. In agreement, Sis1 and the CbpA$^{JD}$-DNAJB1$^{CTD}$ chimera that could establish bidirectional JD–CTD cross tethering with DNAJA2 (*Figure 4—figure supplement 2D* and *Figure 5—figure supplement 1B,C*) due to compatible JD-CTD contact sites, were able to synergize protein disaggregation (*Figure 4—figure supplement 2C* and *Figure 5—figure supplement 1E*). The strength of the JD-driven intermolecular bidirectional tethering correlates with the degree of disaggregation synergy observed with the respective J-protein pairs (*Figure 5—figure supplement 1D,E*). These findings indicate that further refinements have occurred in the electrostatic potentials of eukaryotic JDs to maximize cooperation with opposite class members.

Finally, we provide proximity ligation assay derived biological evidence to confirm our biochemical findings (*Figure 5E–L*). In cultured HeLa cells, V5 tagged DNAJB2 and DNAJB8 did not form red puncta when tested for complex formation with DNAJA2, even under overexpression conditions (*Figure 5G,H*), but readily interacted with Hsp70 via JDs (*Figure 5K,L*). Controls with V5-DNAJB1 displayed complex formation with both DNAJA2 and Hsp70 indicating that the N-terminal tag does not interfere with JD mediated interactions (*Figure 5F,J*). In essence, non-native interactions among members within the J-protein network are simply abolished by an ensuing charge reversion (negative to positive) at the CTD interaction region of J-domains.

## Discussion

In this study, we provide structural, phylogenetic, biochemical and cell biological evidence to support a eukaryote-specific occurrence of interclass complexes between the canonical J-proteins of classes A and B. The distinct change in J-protein biology at the prokaryotic-to-eukaryotic split, where class members form cooperative networks (*Figure 6*), may have triggered functional consequences linked to specific changes in organismal physiology. Habitat wise, bacteria and yeast are exposed to constantly changing harsh environmental stresses such as extreme heat, and are particularly dependent on potent protein disaggregases for stress recovery and survival (*Sanchez and Lindquist, 1990*; *Mogk et al., 1999*). The bacterial ClpB-DnaK system may rely on the broad (size-based) aggregate targeting ability of DnaJ and CbpA for high disaggregation efficiency, which excludes the need for interclass J-protein complex formation. There is also a biological pertinence to the absence of interclass J-protein cooperation in bacteria. DnaJ and CbpA show a clear temporal and spatial separation in expression patterns and intracellular localization in *E. coli* (*Azam et al., 2000*; *Cosgriff et al., 2010*; *Li et al., 2014*; *Yamashino et al., 1994*). This conceivably prevents considerable encountering events between the two molecules in a biological setting hence reducing the chance to evolve a cooperative function. Additionally, the operon linked, J-protein inhibitor CbpM may further contribute to this by stably binding to the JD of CbpA (*Sarraf et al., 2014*) and blocking the evolution of JD-mediated interactions (*Chenoweth et al., 2007*). Moreover, as opposed to eukaryotes, bacteria contain only single copies of the canonical J-protein class members optimized for essential biological functions, which may disfavor positive selection for new features (*Ohta, 2000*; *Richard and Yvert, 2014*) such as sites for opposite class JD interaction. Together, the negative

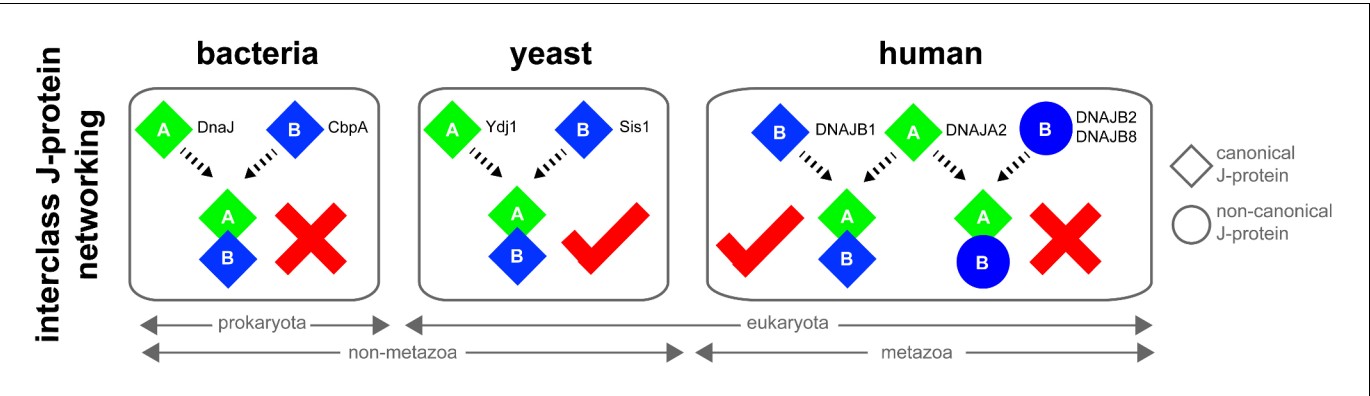

**Figure 6.** Canonical class A and class B members form interclass J-protein networks in eukaryotic cells. Cytosolic yeast and human canonical class A and class B members (e.g. Ydj1 and Sis1, *Saccharomyces cerevisiae*; DNAJA2 and DNAJB1, *Homo sapiens*) form interclass J-protein complexes via complementary binding interfaces at the JDs and the hinge regions of CTDs. *E. coli* cytosol contains only a single pair of canonical class A (DnaJ) and class B (CbpA) J-proteins. These bacterial J-proteins however fail to form interclass J-protein complexes due to the lack of complementary structural features required to establish intermolecular JD-CTD contacts.

selective pressure, the distinct cytosolic localization of DnaJ and CbpA, and the presence of JD blocking CbpM explain why the two J-protein types may not have evolved to interact in *E. coli*.

In contrast, further diversification of eukaryotic class A and B members has resulted in narrowing the substrate selection for the yeast disaggregase system. To compensate for this loss in functional plasticity, we observe the emergence of interclass J-protein complexes that combine different substrate-binding modules of both classes A and B for broader recognition of certain aggregate types. This substrate targeting modulation of eukaryotic disaggregases mediated by J-proteins may have helped evolve new beneficial functions that are absent in prokaryotes; For example, a specialized role of the Hsp104-Ssa1 system in regulating prion-like conformational behavior of naturally occurring cell signaling proteins in unicellular yeast (*Newby and Lindquist, 2013*). In multicellular organisms, additional evolutionary constraints and fitness costs linked to maintaining the Hsp104 system may have triggered the loss of Hsp104 and selected for Hsp70-based disaggregases (*Nillegoda and Bukau, 2015*).

How the different classes of J-proteins (single and in complex) recognize distinct aggregate populations is unclear. The packing of the aggregate, the amino acid composition of the exposed protein segments of trapped molecules, and perhaps even the shape of the binding surface could influence aggregate selection by J-proteins. The J-protein class-specific differences in the substrate-binding domains (*Fan et al., 2005*, *2004*; *Reidy et al., 2014*), interdomain communication (*Reidy et al., 2014*) and binding modes (*Terada and Mori, 2000*) could add further layers of complexity for this function. Our molecular understanding of how J-proteins interact with protein substrates remains largely enigmatic.

In addition to broadening the substrate targeting, interclass J-protein complex formation could also help nucleate Hsp70 oligomerization. This is envisioned to facilitate the assembly of the metazoan Hsp70-based disaggregases on aggregate surfaces (*Nillegoda et al., 2015*; *Nillegoda and Bukau, 2015*). In yeast, such J-protein-mediated conglomerations of Hsp70 may aid in both efficient recruitment and activation of Hsp100 hexamers (*Seyffer et al., 2012*; *Carroni et al., 2014*) on different aggregate types. To achieve a similar outcome, the bacterial disaggregase system seems to employ J-proteins with broad aggregate recognition and is proposed to rely on homo J-protein oligomerization (*Celaya et al., 2016*). The precise basis of J-proteins binding to aggregates has not been defined, but presumably J-proteins nucleate where looped-out polypeptide stretches are available for interaction.

The discriminatory electrostatic potential signals for partner protein selection in J-protein networks, which fine-tune the entire Hsp70-based protein folding system, are communicated by a remarkably simple domain topology of four α-helices in J-domains. The distinct positively charged patch on α-helix II proximal to the HPD motif (*Figure 1C*) facilitates Hsp70 binding (*Lu and Cyr*,

*1998*; *Hennessy et al., 2005*). This interaction may be further modulated in some Hsp70-J-protein pairs for specific functions through the dominantly negatively charged electrostatic cloud next to the HPD motif (e.g. see JDs of DnaJ (*A. aceti*, *Sphingomonas sp.* and *C. ultunense*), Ydj1 (*S. cerevisiae*) and DNAJB8 (*H. sapiens*); *Figure 5D*, *Figure 1—figure supplement 1B* and *Figure 1—figure supplement 2B*). Similarly, the electrostatic patch formed at the opposite end (α-helices I and IV) discriminates J-protein partnering for specialized functions in eukaryotic networks. For the interaction with canonical class A members (e.g. DNAJA2), a negatively charged patch proximal to α-helices I and IV of class B J-domains is required. A charge reversal at this site deters JD-CTD contacts with DNAJA2's CTD, providing specificity to interclass J-protein pairing. It is very likely that this rule of interaction for complex formation is conserved between canonical class A (e.g. DNAJA1, DNAJA2, DNJ-12, Ydj1) and class B members (e.g. DNAJB1, DNAJB4, DNJ-13, Sis1) that show a high degree of electrostatic potential conservation at the JD-CTD contact interfaces (*Figure 1—figure supplement 1* and *Figure 1—figure supplement 2*). On the contrary, a charge reversal at the CTD-binding interface of JDs helps DNAJB2 and DNAJB8 to avoid interacting with DNAJA2 and possibly other class A members. DNAJB6, similar to its homolog DNAJB8, has also lost the bipolar electrostatic potential in the JD suggesting that this non-canonical J-protein may also fail to complex with DNAJA2-like members (*Figure 5—figure supplement 1G*). Our predictions, however, do not fully exclude the possibility that these non-canonical J-proteins are unable to interact with other canonical J-proteins under different growth conditions for joint chaperone actions in eukaryotic cells. In yeast, the J-domain of Tim14 (class C) forms a complex with the pseudo J-domain of Tim16 (class C) during mitochondrial protein import (*Mokranjac et al., 2006*) and such JD-JD driven interactions may also exist among some J- or J-like proteins for specialized functions. Extensive biochemical and functional genomic approaches are now needed to fully understand the extent and regulation of this intricate J-protein network, especially in humans where J-protein targeted chaperone machineries are implicated in a wide range of pathologies including cancer, neurodegeneration, muscular atrophies, metabolic disorders and infectious diseases (*Koutras and Braun, 2014*; *Gibbs and Braun, 2008*; *Knox et al., 2011*; *Maier et al., 2008*; *Synofzik et al., 2014*).

In summary, our broad survey of J-proteins across kingdoms of life captures a eukaryote-specific, adaptive evolution in canonical class A and class B J-proteins to allow for interclass complex formation that modulates Hsp70 machinery targeting. Our data shows how J-proteins, individually or in complex, are employed to regulate the operational efficacy of bacterial, yeast and human disaggregation systems. We finally explain a naturally occurring elegant strategy to correctly pair J-proteins for specialized tasks (e.g. protein disaggregation), especially in humans where over 50 isoforms of J-proteins exist. In effect, the diverse members of eukaryotic class A and B J-proteins deriving from multiple gene duplications are reconnected via selective complex formation to ensure fine-tuning of distinct biological functions.

## Materials and methods

### Sequence extraction and pre-processing

Multiple sequence alignments (MSA) for class A and B J-proteins were built separately as follows: for each class, we collected a seed with curated and manually annotated sequences and aligned them using MAFFT (*Katoh et al., 2002*). Sequences in the class B J-protein seed comprised the J-domain, the GF region and both CTDs. Sequences in the class A J-protein seed additionally contained the characteristic zinc-finger domain. We then used HMMER (*Finn et al., 2011*) to build a hidden Markov model and scan the union of the Uniprot and Swissprot databases to extract homologues for both sub-families. The retrieved sequences were then filtered by removing all hits containing more than 20% gaps. To ensure that no class A sequences were present in the class B alignment (and vice-versa), we further filtered the two datasets as follows: all complete and unaligned sequences from the two MSAs were retrieved; from the class B MSA, we discarded all instances whose complete sequences contained the characteristic zinc-finger (ZF) CxxCxGxG motif. While the canonical ZF typically has four of these motifs, we observed that some ZF domains had variations at one of the two glycines. As a consequence, we only kept sequences in the class A MSA whose complete sequences contained at least two of these characteristic motifs. This procedure resulted in 12215 class A J-protein sequences and 4194 class B J-protein sequences.

## Direct coupling analysis

Direct Coupling Analysis (DCA) was performed using the asymmetric version of the pseudo-likelihood method (*Morcos et al., 2011*; *Ekeberg et al., 2013*). Sequences were reweighted using a maximum of 90% identity threshold (*Ekeberg et al., 2013*; *Hopf et al., 2014*; *Balakrishnan et al., 2011*). For the prediction of inter-class interactions, the interacting sequences of the two classes should be concatenated for each organism. The canonical approach for prediction of protein-protein interactions using DCA consists in using information on the genomic location of the sequences to predict which pairs of sequences are most likely to be paired in an organism. This strategy has been effective in the case of bacterial sequences organized in operons (*Hopf et al., 2014*; *Ovchinnikov et al., 2014*; *Feinauer et al., 2016*), but fails when applied to eukaryotes or bacterial sequences that are distant on the genome. In this work, we faced six issues with the pairing problem: (1) DnaJs are present as multiple paralogs in all organisms. (2) The number of DnaJs strongly varies between organisms. (3) There is no systematic knowledge of interacting partners. (4) DnaJs are not located in operons or on nearby positions in the genome. (5) The systematic matching of all DnaJAs with all DnaJBs in an organism leads to a very large number of possible combinations, most of which probably do not interact. (6) Matching too many non-interacting pairs dilutes the coevolutionary inter-protein signal.

Because of the these difficulties, we adopted the following matching strategy: for each class A J-protein sequence in a given organism, we matched it with a single randomly chosen class B J-protein sequence of the same organism. We also enforced that any class B J-protein sequence was only matched with a single class A J-protein sequence. These class A/B matched sequences are then collected for all species and form a randomly matched MSA. This process was repeated 300 times, resulting in an ensemble of different MSAs. DCA was then performed for each alignment. Finally, we considered those predicted DCA pairs which appeared in at least 5% of the 300 MSAs, as selected using a threshold of 0.8 (*Hopf et al., 2014*) on the normalized DCA score above which inter-protein residue pairs were considered statistically significant.

## Generation of phylogenetic trees

Phylogenetic trees were built using the RaxML software suite (*Stamatakis, 2014*). To decrease their size, MSAs for class A and class B proteins were first pruned, retaining only sequences with maximum 90% identity. Phylogenetic trees were then computed with a standard protocol (20 maximum likelihood searches, 100 bootstraps) and the best tree was returned. We observed that support values (*Yang and Rannala, 2012*) were generally low for the interior branches, but the overall phylogenetic separation was robust, reproducing a coherent phylogeny at large scale.

## Phylogenetic discriminant analysis

We developed a methodology to assess the residues responsible for the phylogenetic and functional differences observed in J-proteins, that we call Phylogenetic Discriminant Analysis (PDA). This method bears some resemblance with the critical variable selection methodology introduced in *Grigolon et al., 2016*) but relies on phylogenetic annotations and thus falls into the category of supervised learning. Our approach was as follows:

For each of the $\binom{N}{3}$ possible position triplets (N is the MSA width), we built a reduced MSA consisting only of these three positions. We then performed principal component analysis (PCA) on this reduced MSA to project the sequences on a maximum-variance subspace (*Casari et al., 1995*). The sequences were then clustered together in this subspace using hierarchical clustering (*Murtagh and Contreras, 2012*). To define the number of clusters, we set a cut-off on the distance between clusters equal to the average distance between all sequences. By doing so, the number of clusters does not need to be explicitly chosen for each position triplet. The homogeneity of each cluster c with respect to phylogeny was then measured by means of the Shannon entropy

$$h(c) = -\sum_{i_c} P(i_c) \log P(i_c)$$

where $P(i_c)$ is the fraction of sequences in cluster c belonging to the phylogenetic group i. The

distributions of phylogenetic groups were then measured for each cluster, and their entropies computed. The average entropy over all clusters is then

$$H(C) = -\sum_{c \in C} w(c)h(c)$$

where w(c) is the fraction of sequences in cluster c and C is the set of all clusters found by the algorithm.

The average entropy over the clusters is then a measure of how well the position triplet discriminates the different sub-classes. If the average entropy is low, then the clusters built on a given triplet contain a lower mixing of phylogenetic classes, and the position triplet is hence a good discriminator of phylogeny. As the entropy is a monotonic function of the 'mixing', the entropy H is thus a good scale to score the discriminative power of triplets: The lower H is, the less mixed are the clusters based on a given triplet of positions.

By computing the distribution of entropies over all triplets (*Figure 2—figure supplement 2*, left sub-panels), we could evaluate an empirical p-value assessing the statistical significance level of a triplet in discriminating the phylogenetic groups. The p-value of a triplet was simply defined as the fraction of triplets having equal or lower entropy score H. The choice of considering triplets was based on experimental evidence showing that a triple RRR mutant in the J-domain abolished cooperation between class A and B J-proteins (*Nillegoda et al., 2015*). We therefore wanted to examine if the choice of these three positions was statistically found from sequence analysis. In the case of the analysis of the J-domains, we could systematically test all possible 54740 triplets, whereas for the CTD analysis, we randomly chose a subset of 50000 triplets to limit the computational burden. We performed sixfold cross-validation and verified that this under-sampling was a good approximation. This method could in principle be extended to the analysis of k-mers (k > 3). However, the number of combinations grows exponentially with k (for the JD with 70 positions, this results in ~900,000 4-mers and 12 millions 4-mers, while for the CTD there are already ~ 160 millions 4-mers). An alternative strategy is to look at all 3-mers, and retain positions that appear most often in the strongly discriminating triplets (*Figure 2—figure supplement 2*, right sub-panels). Here, we have followed this strategy. The most discriminating positions (top five in the main text) were selected as the positions that appeared most often in all triplets having lower entropy than the reference RRR triplet at residues 4, 69 and 70 in DNAJB1 (or 6, 61, and 64 in DNAJA2). The reference RRR triplet was here used as a threshold for considering strongly discriminant positions, as we wanted to test whether this particular triplet had any statistical significance from a phylogenetic discrimination point of view. In the case of the J-domains, the reference triplet had a p-value of 4.6%. For the CTD analysis, where no reference triplets were available, we set the reference p-value to 5%.

In order to set a threshold to select the most frequently appearing residues in the high ranked triplets, we considered a uniform prior null model: If $m$ denotes the number of selected high ranked triplets, the null model for the average probability of each residue to be selected is simply given by $p_{Null}(i) = 3/N_{pos}$ (where $N_{pos} = 70$ for the JD, 254 (resp. 275) for CTD$^B$ (resp. CTD$^A$) (*Figure 2—figure supplement 2*, right sub-panels, dashed red lines). Given a finite sampling of $m$, the standard error of the mean of the null model is given by $\sigma_p = \sqrt{\frac{p_{Null}(1-p_{Null})}{m}}$. This allows estimating a p-value for the outliers of the distributions of selection probability in terms of standard deviations from the mean (*Figure 2—figure supplement 2*, righ sub-panels. Dashed magenta (resp. green) lines denote 3 (resp. 10) standard errors of the mean. We note that the strong assumptions of the null model (completely uniformly distributed triplets), results in outliers having high deviations from the mean (or alternatively very low p-values). In the main analysis, we have used a conservative choice, considering outliers above 10-sigma deviations as significant (p-value<$10^{-23}$). We observe that taking a less restrictive selection (three sigma, p-value<$1.5 \, 10^{-3}$) results in the selection of a small number of additional PDA residues, which lie in close vicinity to the ones selected by the more stringent 10-sigma criterion.

To assess the robustness of the PDA analysis, we tested this methodology with different separations of phylogenetic classes (from 'Bacteria-Eukaryotes' up to 'Fungi-Proteobacteria-Firmicutes-Viridiplantae-Other Bacteria-Other Eukaryotes' and found the results to be robust (*Figure 2—figure supplement 2A–G*). Furthermore, we verified that when using another clustering method

(modularity based clustering [*Granell et al., 2011*]), the results did not change (*Figure 2—figure supplement 2H*).

## Protein structure preparation

The same structures and models for human (DNAJA1, DNAJA2, DNAJB1, DNAJB4) and *C. elegans* (DNJ-12, DNJ-13) proteins as in *Nillegoda et al. (2015)* were used. For the other proteins studied, three dimensional structures were available and used for the following: J-domains of Sis1 (PDB ID: 2o37), *E. coli* class A (PDB ID: 1xbl), *E. coli* class B (PDB ID: 3ucs), human DNAJB8 (PDB ID: 2dmx), human DNAJB2 (PDB ID: 2lgw) and CTD of Sis1 (PDB ID: 1c3g) and Ydj1 (PDB ID: 1nlt, 1xao). The structure of the CTD dimer of Ydj1 was built using the structure of the CTD monomer (PDB ID: 1nlt), which was superimposed twice on a crystal structure containing the dimerization site (PDB ID: 1xao) by using PyMOL (http://www.pymol.org). For the remainder of the proteins studied, no crystal or NMR structure was available. Therefore, three-dimensional structures of the domains of these proteins were built by comparative modeling using the Swiss-Model webserver (http://swissmodel.expasy.org) (*Biasini et al., 2014*).

For all class A CTD models, the CTD dimer model of Ydj1 was used as a template structure and the two $Zn^{2+}$ ions were transferred afterwards. For the class A CTD of *Pseudomonas oryzihabitans* (Uniprot accession: A0A0D7F716), a less conserved loop close to the $Zn^{2+}$ binding region was modeled with different backbone coordinates from the template structure but these prohibited realistic $Zn^{2+}$ binding because of a too large binding distance. Therefore, three residues were changed in the sequence to force the Swiss-Model algorithm to model the same backbone coordinates as in the template structure ('KIIPEP' → 'DIIKDP'). Afterwards, the three residues in the model structure were back-mutated using the mutagenesis tool in the PyMOL software. This model was then used as a template structure in the Swiss-Model webserver to slightly adapt the side chains in the mutated region. Only in the case of the *Sphingomonas sp* (strain SKA58) DnaJ (UniProt accession: Q1NCH5) was the model of *Acetobacter aceti 1023* DnaJ (UniProt accession: A0A063 × 4A7), which was built using the Ydj1 model, used as a template because the less conserved loop around the $Zn^{2+}$ binding region was modeled better for $Zn^{2+}$ ion binding than when the Ydj1 model was used. In the case of the *Bordetella pertussis* (UniProt accession: Q7VVY3) class A CTD, the sequence alignment was manually adapted to enable the modeling of the C-terminal region. For this purpose, a multiple sequence alignment of the four gamma and the beta bacterial sequences and the yeast sequence (template structure) was considered using the software DeepView (*Guex et al., 2009*). A DeepView project with the adapted alignment was uploaded to the Swiss-Model webserver.

For the class B CTDs, the Swiss-Model webserver was used to find a template structure and, if multiple templates were found, the one with the highest sequence identity to the target structure was chosen and then, in the case of more than one structure for this sequence, the corresponding structure with the highest QMEAN4 score. For the following class B CTDs (UniProt accession numbers), the template structure 3lz8.B was used: P36659, P63262, W9BQH2, J7RE62, F4JY55. The PDB ID 3lz8.A was used as a template structure for the following class B CTDs (UniProt accession numbers): A0A0D7FE35, Q1NEX3, M1ZLZ3, O75953. For the Type B CTD of A0A063XA16, the structure with the PDB ID 4j80.A was used as a template. The dimer structure of Sis1 was built by superimposing the crystal structure of the monomer (PDB ID: 1c3g) twice on the 19 C-terminal residues of the crystal structure of the JB1 dimer (PDB ID: 3agz).

For the class A J-domain of *Acetobacter aceti 1023* DnaJ (UniProt accession: A0A063 × 4A7), the structure with the PDB id 4j80 was chosen, and for the *Sphingomonas sp* (strain SKA58) DnaJ (UniProt accession: Q1NCH5) and the ATJ3 (UniProt accession: Q94AW8), the structure with PDB ID 4rwu was chosen as the template. In the case of DNAJA4 (UniProt accession: Q8WW22), the structure with PDB ID 2lo1 was taken as the template. For all other class A J-domains, the structure with the PDB ID 1xbl from *E. coli* was used as the template. For the class B J-domains, the following templates were used (UniProt accession: PDB ID of template structure): A0A063XA16:4j7z, Q1NEX3:2dmx, M1ZLZ3:2yua, F4JY55(At5g25530):2m6y, O75953(DNAJB5):4wb7, O75190 (DNAJB6):4j7z. For all other class B J-domains, the *E. coli* structure with the PDB id 3ucs was used.

The structures were prepared by adding polar hydrogen atoms to the protein structures with WHATIF5 (*Vriend, 1990*). The electrostatic potential of each protein was calculated by numerically solving the linearized Poisson–Boltzmann equation with UHBD (*Madura et al., 1995*). Electrostatic potential grids with $250^3$ grid points with 1 Å spacing were used for all proteins. The relative

dielectric constants of the solvent and the protein were set to 78.0 and 4.0, respectively, and the dielectric boundary was defined by the protein's van der Waals surface. The ionic strength was set to 50 mM at a temperature of 300 K, with an ion exclusion radius (Stern layer) of 1.5 Å. The protein atoms were assigned OPLS atomic partial charges and radii (*Jorgensen et al., 1996*).

All class A J-domain structures were superimposed on the DNAJA2 J-domain. The class A CTDs were superimposed on the lower CTD-II domain of DNAJA2. All class B J-domain structures were superimposed on the DNAJB1 J-domain. The class B CTDs were superimposed on the upper CTD-I domain of DNAJB1. All structures were superimposed with the alignment tool of the PyMOL software.

The similarity of the calculated electrostatic potentials of the superimposed structures was computed using the PIPSA (Protein Interaction Property Similarity Analysis) software (*Wade et al., 2001*). The resulting distance matrix was used for a Ward's clustering. Only for Type A CTDs was an average-clustering used, but this yielded similar results to the Ward's clustering. For the local PIPSA analysis, a center and a radius were defined as follows. For the local PIPSA analysis of the class A CTD, the midpoint between the residue K226 in the DNAJA2 CTD and K21 in the DNAJB1 J-domain was chosen. This pair of residues was found in a lysine-specific cross-linking experiment (*Nillegoda et al., 2015*). A docking simulation of the DNAJA2 CTD and the DNAJB1 J-domain supported the domain interaction and the coordinate of the midpoint was taken from the representative complexed structure (see [*Nillegoda et al., 2015*] for more information). The radius of the sphere was set to 25 Å to include the whole predicted interaction site. The same procedure was applied for the DNAJB1 CTD and the DNAJA2 J-domain, for which two cross-linking residues, K209 in the DNAJB1 CTD and the K46 in the DNAJA2 J-domain, were identified. The radius of the sphere was also set to 25 Å. For the PIPSA analysis of the metazoan JDs, average-clustering was applied. All metazoan JD structures were superimposed on the DNAJB1 JD and a sphere with a radius of 25 Å was set to cover the region around α-helix I and IV and the RRR mutation site of DNAJB1$^{RRR}$.

For the Brownian Dynamics simulations, the SDA software (*Martinez et al., 2015*) was used with the same conditions and the same clustering procedure for the docked J-domain as described in our previous study (*Nillegoda et al., 2015*). The docked cluster representatives were used to calculate the average Euclidean distance between their center of geometry and the center of geometry of the previously docked JD$^{DNAJB1}$ (cluster one and two) to the CTD$^{DNAJA2}$ (*Nillegoda et al., 2015*). The CTD of *E. coli* used for the docking simulations was superimposed on the CTD$^{DNAJA2}$ before carrying out the docking simulations. Because of the dimeric structure, the distance to both cluster representatives 1 and 2 was calculated and the smaller distance was used for calculating the average distance.

## Recombinant proteins

Bacterial, yeast and human recombinant proteins were expressed and purified as described previously (*Rampelt et al., 2012*; *Nillegoda et al., 2015*; *Westhoff et al., 2005*; *Haslberger et al., 2008*). The plasmid for His-tagged DNAJB2a purification was obtained from Dr. M. E. Cheetham (University College London, UK).

## Cell culture and growth conditions

*E. coli* strain K-12 MG1655 encoding CbpA-mCherry (*Chintakayala et al., 2015*) was a kind gift from David Grainger (University of Birmingham, UK). Strain NA01 was generated by transforming MG1655 encoding CbpA-mCherry with plasmid pDK194 carrying DnaJ-YFP under T$_7$ promoter. IPTG induction of DnaJ-YFP was carried out as described previously (*Winkler et al., 2010*). Strains were grown in Luria-Bertani (LB) medium at 30°C with appropriate antibiotic selections.

*S. cerevisiae* strains were grown in yeast extract-peptone-dextrose (YPD) media at indicated temperatures using standard methods. Log phase cultures in YPD media obtained at 30°C. Sis1 depletion strain *tet07-sis1* (*MATa; his3-1; leu2-0; met15-0*; pSIS1::*kanR-tet07-TATA URA3::CMV*-tTA) was obtained as kind gifts from Dr. D. Cyr (University of North Carolina, USA). For Sis1 depletion, *tet07-sis1* cells (control: *tet-off* cells) were grown overnight in YPD media, diluted back to OD600 = 0.05 in YPD containing 10 µg/ml doxycycline and grown for 20 hr. Log phase cells for experiments were obtained by diluting cells back to OD600 = 0.05 and allowing three cells divisions in fresh

doxycycline containing media. *ydj1* deletion strain (VCY010, *MATα; his3Δ1; leu2Δ0; lys2Δ0; ura3Δ0; ydj1Δ*::kanMX4) was derived from BY4742.

HeLa cells were obtained from American Type Culture Collection (ATCC-CCL2; RRID:CVCL_0030) were grown in DMEM (Gibco, Thermo Fisher Scientific, UK) supplemented with 10% (v/v) FBS at 37°C in 5% $CO_2$. Mycoplasma contamination of the HeLa cell culture was tested negative with Look-Out Mycoplasm Detection kit (Sigma-Aldrich; MP0035, MO, USA). Plasmids pcDNA5/FRT/TO V5-DNAJA2, V5-DNAJB1, V5-DNAJB2a and V5-DNAJB8 were kind gifts from Harm Kampinga (University of Groningen). Plasmid transfections were carried according to standard protocols using Lipofectamine 2000 (Thermo Fisher Scientific, UK). siRNA transfections were performed according to standard protocols using DharmaFECT one transfection reagent (Dharmacon, CO, USA). HeLa cells were transfected with 50 µM siRNA smartpools (Dharmacon onTarget Plus) against DNAJA2 (Dharmacon, L-012104–01), DNAJB1 (Dharmacon, L-012735–01) or scrambled non-targeting siRNA (Dharmacon, D-001810–10) for 72 hr.

## Duolink in situ proximity ligation assays

Proximity ligation assay in bacteria: Log and stationary phase (grown for 18 hr) *E. coli* cells grown in LB medium at 30°C were fixed by adding ice cold 99% methanol and incubating at −20°C for 30 min. The fixed cells were attached to Poly-L-Lysine coated slides and treated with Lysozyme solution (2 mg/ml Lysozyme, 25 mM Tris HCl pH 8.0, 50 mM glucose, 10 mM EDTA) at room temperature for 30 min. The cells were then washed 3x in 100 ml PBST (140 mM NaCl, 2 mM KCl, 8 mM $K_2HPO_4$, 1.5 mM $KH_2PO_4$, 0.05% tween20). The cells were treated with 99% methanol followed by an acetone wash. Methanol fixation quenched YFP and mCherry fluorescence allowing us to use 561 nm solid-state laser to specifically image signal from DUOLINK fluorophore (orange kit) that hybridize to amplified PCR product. The air-dried cells were then subjected to DUOLINK blocking, antibody treatment (1° antibody dilution, 1:300), ligation, DNA amplification and mounting according to manufacturer's guidelines (Sigma-Aldrich).

Proximity ligation assay in yeast: Log phase *S. cerevisiae* cells grown in YPD medium at 30°C were fixed with 4% para-formaldehyde for 15 min at room temperature and washed 2x with 100 mM $KPO_4$ pH 6.5 buffer and 1x with wash buffer (1.2 M Sorbitol in 100 mM $KPO_4$ pH 6.5). Cell walls were digested with Zymolase solution (500 µg/ml Zymolase 100T, 1.2 M Sorbitol, 100 mM $KPO_4$ pH 6.5, 20 mM 2-Mercaptoethanol) at 30°C for 20 min. The resulting spheroplasts were washed 3X with wash buffer and attached to Poly-L-Lysine coated slides. The attached spheroplasts were then washed 3x with permeabilizing solution (1% TritonX100 in 100 mM KPO4 pH 6.5). DUOLINK blocking, antibody treatment (1° antibody dilution, 1:300), ligation, DNA amplification and mounting steps were carried out according to manufacturer's guidelines (Sigma-Aldrich).

Proximity ligation assay in Hela cells: HeLa cells were grown in DMEM (Gibco) supplemented with 10% (v/v) FBS at 37°C in 5% $CO_2$. Cells were plated at a density of $2 \times 10^4$ cells/well in poly-lysine coated 10-well diagnostic slides (Thermo scientific, MA, USA) and incubated for 24 hr. Cells were fixed with 4% para-formaldehyde in PBS and the proximity ligation assay was carried out according to DUOLINK manufacturer's guideline for mammalian cells (Sigma-Aldrich). A 1:150 1° antibody dilution was used.

## Confocal microscopy

Confocal microscopy was performed on a LSM 780 system (Carl Zeiss, Germany). Images of HeLa cells were taken with 20x/0.8 NA Plan Apochromat objective (Carl Zeiss) and identical acquisition settings with a pinhole of approx. one airy unit. A 63x/1.4 NA Plan Apochromat objective (Carl Zeiss) was used for yeast and bacterial cell imaging. DNA-stained DAPI was excited with a 405 nm pulsed diode laser and the DUOLINK signal was excited with a 561 nm solid-state laser.

## Antibodies

Commercially available antibodies against DNAJA2 (rabbit monoclonal), DNAJB1 (mouse monoclonal), V5 tag (mouse monoclonal), GAPDH (mouse monoclonal), Pgk1 (mouse monoclonal) and mCherry (mouse monoclonal) were obtained from Abcam (UK)(ab157216; RRID:AB_2650527, Enzo life sciences (NY, USA) (ADI-SPA-450-E; RRID:AB_10621843), Invitrogen (CA, USA)(R960-25; RRID: AB_2556564), Sigma (G8795; RRID:AB_1078991), Invitrogen (459250; RRID:AB_2532235) and

Abcam (ab125096; RRID:AB_11133266), respectively. Anti-mouse Ydj1 (SMC-150; RRID:AB_ 2570364) and anti-rabbit Sis1 (COP-080051; RRID:AB_10709957) were obtained from StressMarq Biosciences Inc. (Canada), and Cosmo Bio Co. (Japan), respectively. Antibody against YFP (rabbit polyclonal; RRID: AB_2650530) was generated in the laboratory. Anti-DnaK (rabbit polyclonal; RRID: AB_2650528) and anti-human Hsp/Hsc70 (rabbit polyclonal; RRID:AB_2650529) antibodies were a kind gift from Matthias Mayer (University of Heidelberg).

## Western blotting

Western blot analysis was carried out using standard methodologies. Alkaline phosphatase conjugated anti-rabbit (AP-1000; RRID:AB_2336194) and anti-mouse (AP-2000; RRID:AB_2336173) IgG (H +L) antibodies from Vector Laboratories (CA, USA) were used as secondary antibodies. The detection was carried out with ECF (GE Healthcare, IL, USA).

## Luciferase assays and SEC-based aggregate profiling

Luciferase refolding-only, luciferase disaggregation/refolding and size exclusion chromatography were performed as previously described (*Nillegoda et al., 2015*). In brief, protein aggregates were generated by heating 25 nM luciferase (final concentration set to 20 nM) with 125 nM sHSP26 at 45°C for 15 min in HKM buffer (50 mM Hepes-KOH pH 7.5, 50 mM KCl, 5 mM MgCl$_2$, 2 mM DTT, 2 mM ATP pH 7.0, 10 μM BSA). Denatured monomeric luciferase was obtained by heating 20 nM luciferase with 100 nM sHSP26 at 42°C for 10 min in HKM buffer containing chaperones of the indicated disaggregase system. Following concentrations of disaggregase components were used in disaggregation/refolding experiments. Bacterial: 750 nM ClpB, 750 nM DnaK, 250 nM J-protein (total) and 75 nM GrpE. Yeast (non-saturating): 750 nM Hsp104, 750 nM Ssa1, 250 nM J-protein (total) and 38 nM Sse1. Yeast (saturating): 2 μM Hsp104, 2 μM Ssa1, 667 nM J-protein (total) and 100 nM Sse1. Human: 750 nM HSPA8, 250 nM J-protein (total) and 38 nM HSPH2.

## Förster resonance energy transfer (FRET) measurements

DNAJA2/ DNAJA2$^{RRR}$ was labeled with ReAsH and DNAJB1 with Alexa Fluor 488 as described before (*Nillegoda et al., 2015*). FRET measurements were performed for the following FRET pairs: CTD-labeled DNAJB1 together with J-domain-labeled DNAJA2/ DNAJA2$^{RRR}$. Emission spectra were recorded on Jasco FP750 spectrofluorimeter at 30°C and quenching of donor fluorescence (Alexa Fluor 488) was quantified at 517 nm and expressed as percentage of donor fluorescence in the absence of acceptor. Human J-proteins were mixed at 0.1 μM DNAJB1 and 1 μM DNAJA2/ DNAJA2$^{RRR}$ in 25 mM HEPES pH 7.5, 50 mM KCl, MgCl$_2$, equilibrated for 15 min at 30°C. For competition measurements, 5 μM (5-fold excess relative to acceptor protein) of unlabeled bacterial (DnaJ, CbpA), yeast (Ydj1, Sis1), human (DNAJB1, DNAJB2a, DNAJB8, DNAJB1$^{RRR}$) and chimeric J-proteins were added to the aforementioned labeled J-protein pairs and equilibrated for 15 min at 30°C before fluorescence measurements. Experiments were performed in triplicate.

## Acknowledgements

We thank Holger Lorenz and Aliakbar Jafar Pour (ZMBH Imaging Facility, Heidelberg University) for their support with confocal microscopy and image processing. We are grateful to Matthias Mayer for critically reading the manuscript and Vidyadhar Nandana for technical and experimental support. This work was funded by the Deutsche Forschungsgemeinschaft (SFB1036, BU 617/19–3 to BB), Alexander von Humboldt Foundation Postdoctoral Fellowship (NBN), German Federal Ministry of Education and Research (BMBF) Virtual Liver Network (0315749 to RCW), EU FEP Flagship Programme Human Brain Project (604102 to RCW), Klaus Tschira Foundation (AS and RCW), French Agence Nationale de la Recherche (ANR-14-ACHN-0016 to AB) and the Swiss National Science Foundation (http://www.snf.ch/) (2012_149278 and 20020_163042/1 to DM).

# Additional information

## Funding

| Funder | Grant reference number | Author |
|---|---|---|
| Alexander von Humboldt-Stiftung | | Nadinath B Nillegoda |
| Klaus Tschira Stiftung | | Antonia Stank<br>Rebecca C Wade |
| Schweizerischer Nationalfonds zur Förderung der Wissenschaftlichen Forschung | 2012_149278 & 20020_163042/1 | Duccio Malinverni |
| Bundesministerium für Bildung und Forschung | Virtual Liver Network 0315749 | Rebecca C Wade |
| Horizon 2020 | FET Flagship Programme Human Brain Project 604102 | Rebecca C Wade |
| Deutsche Forschungsgemeinschaft | SFB1036 BU617/19-3 | Bernd Bukau |

The funders had no role in study design, data collection and interpretation, or the decision to submit the work for publication.

## Author contributions

NBN, Conceptualization, Formal analysis, Supervision, Funding acquisition, Validation, Investigation, Visualization, Methodology, Writing—original draft, Project administration, Writing—review and editing, Experimental design; ASt, DM, Formal analysis, Validation, Investigation, Methodology, Writing—original draft, Writing—review and editing, Experimental design; NA, Investigation, Design experiments; ASz, Formal analysis, Investigation, Writing—review and editing, Design experiments; AB, Investigation, Methodology; PDLR, Formal analysis, Writing—review and editing, Experimental design; RCW, Formal analysis, Writing—review and editing, Experimental design, Funding acquisition, Methodology; BB, Conceptualization, Formal analysis, Funding acquisition, Writing—review and editing

## Author ORCIDs

Nadinath B Nillegoda, http://orcid.org/0000-0002-9980-3991
Antonia Stank, http://orcid.org/0000-0002-3424-4500
Duccio Malinverni, http://orcid.org/0000-0002-3946-9709
Niels Alberts, http://orcid.org/0000-0001-6963-5475
Anna Szlachcic, http://orcid.org/0000-0002-4590-4375
Alessandro Barducci, http://orcid.org/0000-0002-1911-8039
Rebecca C Wade, http://orcid.org/0000-0001-5951-8670
Bernd Bukau, http://orcid.org/0000-0003-0521-7199

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
