## [Decision Letter]

Thank you for submitting your article "Evolution of an intricate J-protein network driving protein disaggregation by the Hsp70 chaperone machinery" for consideration by *eLife*. Your article has been reviewed by two expert peer reviewers, and the evaluation has been overseen by Jeffery Kelly as the Reviewing Editor and Diethard Tautz as the Senior Editor. The reviewers have opted to remain anonymous.

The reviewers have discussed the reviews with one another and the Reviewing Editor has drafted this decision to help you prepare a revised submission.

Summary:

The existence of a J-protein network that acts via interclass complex formation in both animals and simpler eukaryotic unicellular organisms such as yeast, but not in bacteria for disaggregation is reported. The identification, by phylogenetic and coevolutionary analyses, of the hinge region between CTD subdomains as a primary determinant of interclass JD interaction is also reported, which agrees with previous experimental and simulation data (Nillegoda et al., 2015) and supports the idea that these DNAJs have two distinct interaction sites, one for Hsp70 and one for an interclass J-protein interaction. Experimental data was reported to confirm DnaJ-pair formation between DNAJA2-DNAJB1 (human), Ydj1-Sis1 (*S. cerevisiae*), but not DnaJ-CbpA (*E. coli*) and second DnaJ-DnaJ synergism in reactivating heat aggregated luciferase (or lack thereof for the prokaryotic variants).

Essential revisions:

Figure 4, subsections “J-protein class specific aggregate targeting regulates yeast disaggregase system” and “*E. coli* J-proteins target ClpB-DnaK disaggregase to all aggregate populations independent of J-protein class”: The distinction in the action of single versus combined DNAJ on the basis of aggregate size seems rather crude and not very informative in terms of what the individual versus combined DNAJs recognise as substrates (see also below, Figure 6). Please address.

Figure 4—figure supplement 2, subsection “Cross-species chaperone communication reveals additional specialization of eukaryotic J-proteins for Hsp70-based protein disaggregation”, first paragraph: Especially for panel B, I find it difficult to conclude from the data, as they are presented, whether the yeast + human combinations truly synergise on disaggregation or whether this is merely an additive effect of the actions of the individual human and yeast proteins. This may be a matter of visualisation/plotting the data, but here the authors need to clarify how they come to this conclusion. Also for the FRET competition assay, the effects of the yeast-human combinations seem much less than those for the human-human combination, a finding that the authors do not (but should) discuss (also related to the next point).

Figure 5—figure supplement 1: The CbpAJD-DNAJB1CTD and DNAJB8JD-DNAJB1CTD fusions show indeed reduced interaction with DNAJA2, but the level of quenching is still similar as found for Sis1 or Ydj1 (Figure 4—figure supplement 2). For Sis1 or Ydj1, this level of interaction was concluded to be sufficient for cooperation on luciferase disaggregation, while it was not for the DNAJB8JD-DNAJB1CTD fusion. Yet again, the CbpAJD-DNAJB1CTD did collaborate with DNAJA2 in disaggregation. The authors all notice this, but – to my opinion – do not provide a clear explanation as to why this would be. In my opinion, this would imply that besides the J-CTD interaction, also a CTD-substrate interaction is crucial in the combined action of the J-pairs? Please address.

Figure 6: I find the model not very clear and besides that I think that the interpretation in terms of narrow and broad aggregates being targeted a bit oversimplified. Whereas formally correct in terms of what they find for luciferase aggregates, I find it not very likely that size of aggregates per se is key to the synergy of JA+JB. Rather, I would suggest that different structural determinants (including exposed regions) of the aggregated substrates are key as to whether combined J-actions are required. In addition, I tend to disagree to classify DNAJB2 and DNAJB8 as involved in non-specific network interactions. First of all, the "specificity" of the DNAJB1-DNAJA2 combination is not defined beyond aggregate size only. Second, DNAJB2 and DNAJB8 may have also specificity and also could be involved in an interaction network on other substrates. Finally, what lacks in the model is how this can be extrapolated to the in vivo situation, where – in yeast – Hsp104 in present as powerful disaggregase. So, all in all, I think the model should be withdrawn or drastically revised.

---

## [Author Response]

*Essential revisions:*

*Figure 4, subsections “J-protein class specific aggregate targeting regulates yeast disaggregase system” and “E. coli J-proteins target ClpB-DnaK disaggregase to all aggregate populations independent of J-protein class”: The distinction in the action of single versus combined DNAJ on the basis of aggregate size seems rather crude and not very informative in terms of what the individual versus combined DNAJs recognise as substrates (see also below, Figure 6). Please address.*

SEC in principle separates proteins (monomeric/ oligomeric) in solution by their size. However, this separation may be influenced not only by size, but also by the shape and density (nature of packing) of the protein aggregates detected. Further characterization of these amorphous aggregates is difficult due to the heterogeneity of protein interactions within the aggregates. Additionally, the different aggregate populations separated from SEC have very low concentrations of aggregates (<3 nM for each population; based on monomer concentration), which make these populations extremely difficult to experimentally dissect for unique features. Moreover, changing the concentration of the substrate at which the heat aggregation was performed (20 nM, luciferase) to generate more aggregates is also not possible, because aggregates formed at higher substrate concentrations are characteristically different and show very low solubilization efficiency in our disaggregation assays (Rampelt et al., 2012; Nillegoda et al., 2015). In line with this point, we are unable to describe what the exact different classes of J-proteins (single and in complex) recognize on the different aggregate types in the current work. For instance, aggregate selection could be influenced by the amino acid composition of the exposed protein segments of trapped molecules, the shape of the binding surface and distance of neighboring binding sites, and the binding mode of the J-protein or J-protein complex, adding layers of complexity for this function. Our understanding of how J-proteins select their substrates is still largely enigmatic. Although it is highly interesting to characterize the aggregate binding specificities of single versus combined J-proteins, obtaining experimental evidence to address the issue is extremely difficult and is beyond the scope of this work. However, in response to the reviewer’s request for a better explanation of the phenomena in the text, we have now provided a description of what the single versus combined J-proteins may be recognizing on the aggregate surfaces.

Results:

“To dissect the disaggregation process further and determine aggregate specificities of bacterial and yeast class A and B J-proteins, we employed a size-exclusion chromatography (SEC) based strategy (Nillegoda et al., 2015). […] For simplicity, we define these aggregate fractions by size (F1, ≥5000 kDa; F2, 700 – 4000 kDa; F3, 200 – 700 kDa) (Figure 4). Soluble monomeric luciferase (~63 kDa) elutes in the F4 fraction.”

Discussion:

“What the different classes of J-proteins (single and in complex) recognize on a distinct aggregate population is unclear. […] Our molecular understanding of how J-proteins interact with protein substrates remains largely enigmatic.”

*Figure 4—figure supplement 2, subsection “Cross-species chaperone communication reveals additional specialization of eukaryotic J-proteins for Hsp70-based protein disaggregation”, first paragraph: Especially for panel B, I find it difficult to conclude from the data, as they are presented, whether the yeast + human combinations truly synergise on disaggregation or whether this is merely an additive effect of the actions of the individual human and yeast proteins. This may be a matter of visualisation/plotting the data, but here the authors need to clarify how they come to this conclusion.*

We have added new data (New Figure 4—figure supplement 2) to show that indeed the yeast + human interclass J-protein cooperation reported in Figure 4—figure supplement 2 truly synergizes the Hsp70-based disaggregation activity. We have graphed the reactivation of aggregated luciferase in the presence of the indicated class A and B mixtures. In order to differentiate between additive and synergistic affects, we have measured the activities of individual J-proteins at the concentrations used in the J-protein combination experiments. For example, we measured the disaggregation activity in reactions containing 0.12x Ydj1 + 0.5x DNAJB1 (class A+B mixture, x denotes fold-factor), 0.12x Ydj1 (single class A) and 0.5x DNAJB1 (single class B) and added the two single class reactions (dotted curve) to evaluate synergy vsadditiveaffects. We observe that the single class reactions do not sum up to the level of disaggregation/refolding activity observed in the mixed class reaction, which shows that the yeast and human J-proteins cooperate across species and form class A+B J-protein complexes to synergize protein disaggregation.

This is now indicated in the main text and the figure legend of Figure 4—figure supplement 2.

Results:

“Further, when paired with human DNAJA2 and DNAJB1 to form A+B cross-species J-protein combinations, only yeast (and not *E. coli*) members could cooperate and synergize protein disaggregation (Figure 4—figure supplement 2) well above additive effects (Figure 4—figure supplement 2).”

*Also for the FRET competition assay, the effects of the yeast-human combinations seem much less than those for the human-human combination, a finding that the authors do not (but should) discuss (also related to the next point).*

As described below, it is not surprising that the level of donor quenching by excess unlabeled Ydj1 and Sis1 is less, but significant, compared to the human counterparts, DNAJA2 and DNAJB1 (Figure 4—figure supplement 2). We have discussed this in the revised manuscript.

Results:

“The observed cross-species interactions among yeast and human J-proteins correlate with the general conservation of JD interaction sites on eukaryotic class A and B J-proteins (Figure 1). […] Together, these findings suggest that the communication between class A and class B J-proteins is also important for Hsp70-based disaggregase machine assembly and/or architecture.”

*Figure 5—figure supplement 1: The CbpAJD-DNAJB1CTD and DNAJB8JD-DNAJB1CTD fusions show indeed reduced interaction with DNAJA2, but the level of quenching is still similar as found for Sis1 or Ydj1 (Figure 4—figure supplement 2). For Sis1 or Ydj1, this level of interaction was concluded to be sufficient for cooperation on luciferase disaggregation, while it was not for the DNAJB8JD-DNAJB1CTD fusion. Yet again, the CbpAJD-DNAJB1CTD did collaborate with DNAJA2 in disaggregation. The authors all notice this, but – to my opinion – do not provide a clear explanation as to why this would be. In my opinion, this would imply that besides the J-CTD interaction, also a CTD-substrate interaction is crucial in the combined action of the J-pairs? Please address.*

We have performed two new experiments to address the reviewer’s concern (See new Figure 4—figure supplement 2 and new Figure 5—figure supplement 1). Together with the already existing data, we now provide compelling evidence and a clear explanation for our conclusions made in the main text.

Results:

“To test our hypothesis, we generated a DNAJB1 chimera containing either the JD of DNAJB8 or the JD of CbpA (control). The *E. coli* CbpA^JD^ has a charge bipolarity character comparable to eukaryotic class B^JDs^ (Figure 1 and Figure 5—figure supplement 1). […] These findings indicate that further refinements have occurred in the charge bipolarity of eukaryotic JDs to maximize cooperation with opposite class members.”

*Figure 6: I find the model not very clear and besides that I think that the interpretation in terms of narrow and broad aggregates being targeted a bit oversimplified. Whereas formally correct in terms of what they find for luciferase aggregates, I find it not very likely that size of aggregates per se is key to the synergy of JA+JB. Rather, I would suggest that different structural determinants (including exposed regions) of the aggregated substrates are key as to whether combined J-actions are required. In addition, I tend to disagree to classify DNAJB2 and DNAJB8 as involved in non-specific network interactions. First of all, the "specificity" of the DNAJB1-DNAJA2 combination is not defined beyond aggregate size only. Second, DNAJB2 and DNAJB8 may have also specificity and also could be involved in an interaction network on other substrates. Finally, what lacks in the model is how this can be extrapolated to the in vivo situation, where – in yeast – Hsp104 in present as powerful disaggregase. So, all in all, I think the model should be withdrawn or drastically revised.*

In adherence to the reviewer comments we have revised our model and simplified it to only portray the evolution of interclass complex formation among canonical J-proteins of class A and B (New Figure 6).